# HexMachina: Self-Evolving Multi-Agent System for Continual Learning of Catan

## Abstract

We aim to improve on the long-horizon gaps in large language model (LLM) agents by enabling them to sustain coherent strategies in adversarial, stochastic environments. Settlers of Catan provides a challenging benchmark: strategic success depends on balancing short- and long-term goals in the face of dice randomness, trading, expansion, and blocking. This is difficult because prompt-centric LLM agents (e.g., ReAct, Reflexion) must re-interpret large, evolving game states every turn, quickly saturating context windows and failing to maintain consistent strategy across episodes. We propose *HexMachina*, a continual learning multi-agent system that separates environment *discovery* (inducing an adapter layer without documentation) from strategy *improvement* (evolving a compiled player). This architecture preserves executable artifacts, letting the LLM focus on high-level strategy design rather than per-turn decision-making. In controlled Catanatron experiments, HexMachina learns from scratch, evolving players that outperform the strongest human-crafted baseline (AlphaBeta). Our best runs achieve a 54% win rate against AlphaBeta, outperforming prompt-driven LLM agents and shallow no-discovery baselines. Ablations further confirm that greater focus on pure strategy improves performance. Theoretically, this shows that artifact-centric continual learning can transform LLMs from brittle per-turn deciders into stable strategy designers, providing a reusable path toward long-horizon autonomy.

## 1 Introduction

Prompt-centric LLM agents and multi-agent systems are powerful, but they struggle on long-horizon tasks: as episodes unfold, prompts saturate with state summaries and ad-hoc "memory," forcing the model to re-interpret the environment at every step (Aghzal et al. (2025); Nayak et al. (2025); Chen et al. (2024)). To move toward autonomous task following and long-horizon competence, an agent should not have to relearn the interface to their environment at each inference step (Bubeck et al. (2023); Park et al. (2023)). This has motivated experimentation with continual learning agent designs that embed feedback loops and let LLMs revise their own prompts and even generate tools/-code to improve over time (Zelikman et al. (2022)). In particular, letting an agent gather and preserve artifacts (e.g., reusable functions and typed helpers) offloads heavy context parsing to deterministic code so the model can focus on designing strategy, not re-describing the world.

Despite progress in continual learning, there are few benchmarks that test whether agents can refine a coherent strategy over long horizons. Most existing domains emphasize short tasks or broad skill discovery, offering limited insight into how well an agent can sustain and improve a single competitive policy. Yet this ability is crucial: real-world applications often require agents not just to explore or act locally, but to commit to strategies that hold over many steps in the presence of uncertainty and competition. A benchmark that demands persistent strategy refinement against a strong adversary is therefore essential for evaluating whether lifelong agents truly overcome the long-horizon gap.

Settlers of Catan is an ideal stress test: each turn presents a large, evolving state and action space; success depends on balancing short- and long-term rewards under stochastic resource production, trading, expansion, and adversarial play. Using the open-source Catanatron framework (Collazo (2025)) gives us a controlled interface to observe how a lifelong architecture impacts performance in a domain that reliably exposes limits in long-horizon reasoning.

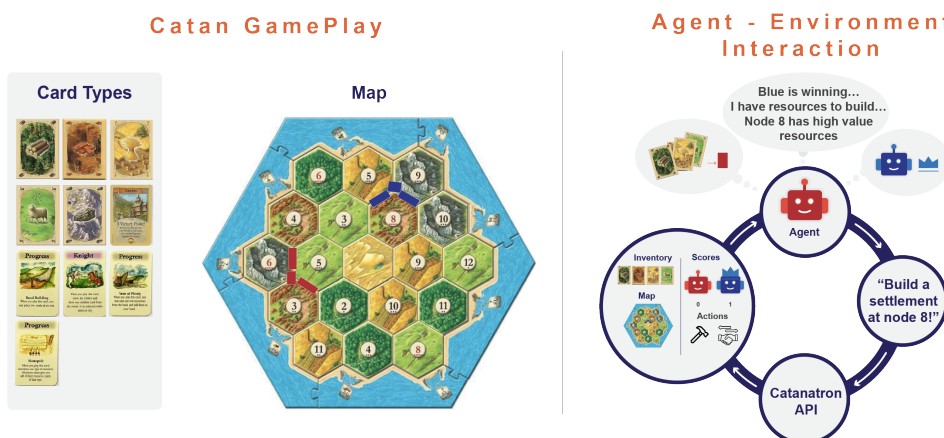

Figure 1: **Overview of Catan gameplay and LLM-agent interaction**. **Left**: *Settlers of Catan* – Players take turns to gather, trade, and spend resources to build on a modular board in a stochastic, partially observable strategy game. The objective is to reach 10 victory points by constructing settlements, roads, and cities Catan Fusion; Catan Collector. **Right**: Our LLM-based framework interacts with the Catanatron API, leveraging game state information and strategic reasoning to decide actions. Through repeated play and self-modification, agents evolve more coherent long-term strategies (Smashicons; murmur (a;b); Hilmy Abiyyu A.; yaicon).

We first demonstrate that traditional per-turn LLM agents (e.g., ReAct/Reflexion-style) perform poorly against a strong human-crafted bot. Asking the model to parse the full game state and independently choose every action while attempting to "hold" a global plan proves unreliable and inconsistent (Table 2). To address this, we separate the act of *thinking* from the act of *playing*, drawing inspiration from the AutoGPT framework (Yang et al. (2023)) to define distinct agent roles: Orchestrator, Analyst, Strategist, Researcher, and Coder. In this configuration, the system hypothesizes a strategy, translates it into a player implementation, reviews the API to ensure correctness, and then evaluates and improves through repeated play. While this Voyager-style (Wang et al. (2023a)) continual learner shows progress, it tends to converge on shallow heuristics that fail to capture the depth of strategic play required in Catan (Appendix A.2). Motivated by this limitation, we introduce a clean separation between the discovery of executable API artifacts and the refinement of strategies built on top of them. With this split, our system, *HexMachina*, evolves players that consistently execute intelligent, long-horizon strategies,outperforming traditional LLM agents, common continual learning architectures, and even the AlphaBeta baseline.

**Main Contributions**. We highlight the following key contributions from our work:

- **HexMachina: Self-Evolving LLM Agent Framework.** An autonomous system that learns an unknown environment without formal documentation, preserves key code/knowledge as artifacts, and improves its strategy via a closed-loop process that generates and executes code with no human intervention.

- **A strong benchmark setting for continual LLM-agent learning:** *Settlers of Catan*. An environment that both requires long-horizon strategy and distracts naive agents with a large, changing state/action space and delayed rewards.

- **Lifelong agents beat traditional LLM agents on Catan.** HexMachina outperforms prompt-driven baselines and rivals the best human-engineered Catanatron bot (AlphaBeta) by letting the LLM design strategy while compiled code executes it consistently.

- **Empirical importance of separating discovery and improvement.** We show that decoupling environment-artifact discovery from strategy refinement materially improves strategy quality and game performance.

Table 1: **Focus comparison.** ✓=yes, ∼=partial, ×=no. Policy evolution (broad: direct or via reward/program/skill search); Artifacts (persisted executable code/skills); Induction (doc-free adapter induction; Voyager ∼ with provided control primitives); Adversarial strategy-based (head-to-head vs strong fixed opponent; ✓ only for HexMachina).

| System | Environment | Induction | Artifacts | Adversary | Evolution |
|---|---|---|---|---|---|
| Voyager | Minecraft | ∼ | ✓ | × | ✓ |
| AlphaEvolve | Code | × | ✓ | × | ✓ |
| Eureka | Isaac Gym | × | ✓ | × | ✓ |
| **HexMachina** | Catanatron | ✓ | ✓ | ✓ | ✓ |

## 2 RELATED WORKS

**Game-Playing AI and Strategy Games** Games have long served as benchmarks for AI research (Gallotta et al. (2024); Costarelli et al. (2024); Nasir et al. (2024)). While significant progress has been made in perfect-information games like Chess and Go (Schultz et al. (2024); Silver et al. (2016)), strategic board games such as *Settlers of Catan*, *Diplomacy* ( (FAIR)) or *Civilization* (Qi et al. (2024)) introduce elements of expanding action spaces, partial observability, and multi-agent interaction, posing unique challenges to an AI system (Szita et al. (2009)). Previous works approached Catan using a specialized neural network architecture to handle its mixed data types, enabling an RL agent to outperform traditional rule-based bots (Gendre & Kaneko (2020)). In contrast, our approach leverages LLMs' natural language understanding to navigate Catan's complexities, focusing on autonomous game-play discovery and strategy refinement without relying on extensive training data.

**LLM Agents and Long-Horizon Planning** LLMs reason well locally but falter at multi-step autonomy: studies report low success on end-to-end plan generation, with models performing better as advisors to external planners Valmeekam et al. (2023). Benchmarks like TravelPlanner confirm poor pass rates even with tools and staged prompting, revealing brittleness under constraint-heavy, multi-objective tasks Xie et al. (2024); Zheng et al. (2025); Nayak et al. (2025); Cui et al. (2025). Prompt-centric agents (ReAct, Reflexion) still act per-turn from ever-growing text context, and multi-agent scaffolds (CAMEL, AutoGen) coordinate via dialogue Yao et al. (2023); Shinn et al. (2023); Wei et al. (2023); Xi et al. (2025); Li et al. (2023); Wu et al. (2023); yet in long-horizon, adversarial domains they repeatedly re-parse large states and lack a persistent executable substrate to enforce strategy across an episode, leaving the planning gap largely intact.

**Self-Improvement and Continual Learning Agents** Inference-time self-improvement spans verbal reflection (Reflexion), evolutionary prompt search (PromptBreeder, PromptAgent), and code-writing agents that iteratively refine programs (Shinn et al. (2023); Fernando et al. (2023); Wang et al. (2023b)). Surveys systematize these inference-time strategies and the broader landscape of LLM agents (Song et al. (2024); Dong et al. (2024)). Eureka (Ma et al. (2024)) explores program and reward evolution, demonstrating how automated search over reinforcement learning environments can uncover novel control strategies. AlphaEvolve (Novikov et al. (2025)) presents an evolutionary coding agent to tackle open scientific problems and algorithm improvement. Embodied lifelong systems like Voyager show that storing executable skills (a skill library) improves persistence and reuse across episodes, but emphasize breadth (discovering many primitives) rather than depth (refining a single competitive policy).

Building on Voyager, Eureka, and AlphaEvolve, which respectively advance skill discovery, reward/program evolution, and automated code improvement, we shift focus to a different question: can a lifelong LLM system, operating without documentation, induce a compact adapter to an unknown environment and persist executable artifacts in order to evolve a single competitive policy that outperforms traditional LLM agents in adversarial play?

## 3 BACKGROUND

**Settlers of Catan as a Strategic Benchmark**  *Settlers of Catan* is a 3-4 player board game where players collect and trade resources to build settlements and roads, racing to earn 10 victory points on a modular island map. The game emphasizes resource management, planning, and negotiation, with mechanics like the robber (which blocks resources) adding tactical depth. Catan is known for its balance of luck and skill. **Victory** goes to the first player to reach **10 points**, earned by building and upgrading settlements into cities, buying development cards, and achieving goals like the longest road or largest army. Each settlement is worth 1 point, each city 2, and some development cards grant hidden points or knight bonuses. **Every turn** starts with a dice roll that produces resources for players with adjacent settlements. The active player may then trade and build. If a 7 is rolled, the robber is activated, blocking a tile and stealing a resource. Players must plan expansions, balance upgrades, and trade strategically to manage luck. This need for adaptation and foresight makes Catan a strong benchmark for evaluating strategic reasoning in agents.

**The Catanatron Framework**  We use the open-source Python-based simulator **Catanatron** as our evaluation environment. Designed for automated gameplay of *Settlers of Catan*, Catanatron offers a programmatic interface for integrating custom agents and supports rapid simulation at scale. It faithfully implements the game's rules and dynamics, capturing key strategic elements such as resource management, trade negotiation via structured proposals, and randomness introduced by dice rolls. Each game consists of players competing to reach ten victory points, with players interacting through well-defined game states that include current resources, board positions, available actions, and observable opponent statuses. Games typically span 40 to 100 turns, allowing for extended observation of agents' long-term planning capabilities. We benchmark our LLM-driven agents against **AlphaBeta**, the best-performing heuristic agent provided through the API which uses a depth-2 alpha-beta pruning algorithm with heuristic evaluation to select actions.

**Alpha-Beta Benchmark**  Our primary baseline is Catanatron's AlphaBeta agent: an alpha-beta minimax over stochastic outcomes that computes the expected value of successor states via chance expansion and a fast heuristic value function. Concretely, it uses a depth-2 search (default), a 20 s decision cap, and an optional action-space pruning mode (e.g., robber and maritime-trade pruning heuristics). At leaves, it applies a parameterized value function, and it short-circuits when only one legal action exists. We adopt the author defaults unless otherwise noted, fixing depth = 2 for all reported comparisons. This baseline is both strong and extremely fast, enabling thousands of head-to-head evaluations needed by our continual-learning setup.

## 4 HEXMACHINA

HexMachina is an autonomous self-evolving multi-agent system that crafts a powerful Catanatron player capable of rivaling the top human-crafted baselines. We utilized Langchain for the model agnostic services, and Langraph for the state machine. Once launched, HexMachina begins by running a *discovery phase*, were it gathers information about the Catanatron API to evolve an **adapters** file. After completion, it enters an *improvement phase* where it begins evolving a **player** file. Each evolution consists of agent collaboration until the Coder writes improvements in the form of testable code. Each phase is limited to 20 evolutions, counted by each time the Coder is called.

### 4.1 CAPABILITIES

Listed below are the capabilities that enable HexMachina to employ continual learning effectively:

**Player Generation**  HexMachina incrementally codes a complete Catanatron **player** module during the *improvement phase*. The process begins with a minimal template that simply returns the first legal action, then evolves into increasingly sophisticated strategies as feedback accumulates. Importantly, the generated player is not just a script of next actions but an executable policy that can consistently carry out a long-term plan across an entire game. This design shifts the LLM's role from being a per-turn decider to being a strategy architect, with the Coder agent ensuring that every idea is grounded in syntactically valid and testable code.

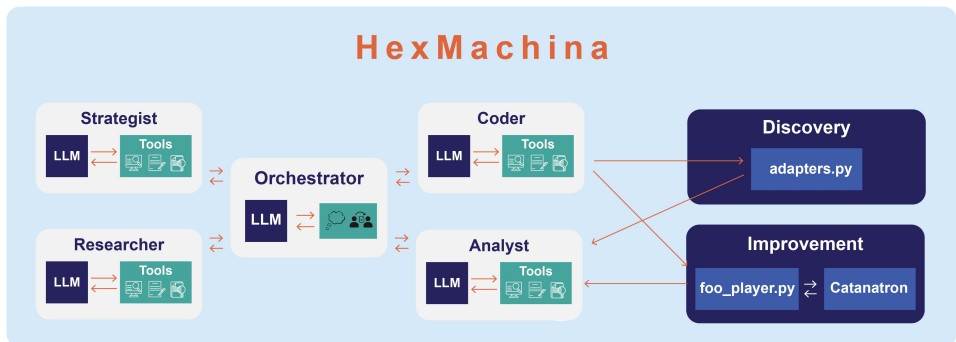

Figure 2: **HexMachina Architecture.** During the discovery phase, the Orchestrator coordinates agents to induce executable functions into the `Adapter` file, stabilizing access to the environment. In the improvement phase, we found it most effective to rely on a streamlined loop of *Analyst*, *Coder*, and *Orchestrator*, avoiding dilution from additional roles. Here, the Analyst diagnoses performance, the Coder translates revisions into executable code, and the Orchestrator manages iteration. This separation enables the system to refine `FooPlayer` into a consistent long-horizon strategy.

**Experimentation Engine** At the core of HexMachina is a deterministic experimentation harness. This engine repeatedly pits evolving players against the strong AlphaBeta baseline under fixed seeds and identical settings, logging outcomes, intermediate states, and decision traces. By holding the environment constant, we can attribute changes in win rate or victory points directly to code evolution, rather than stochastic noise. This repeatable evaluation cycle transforms raw self-play into structured experimental evidence for policy improvement.

**Evaluation** Evaluation closes the loop: after each batch of games, HexMachina analyzes outcomes to identify which strategic choices were beneficial and which led to failure. This feedback is distilled into concise summaries that the Orchestrator uses to decide whether to preserve, modify, or discard a candidate player. In practice, we found that the most effective evaluation loop did not require every agent's perspective; instead, a streamlined pipeline of *Analyst to Coder to Orchestrator* yielded clearer strategic signals. Additional recommendations from other roles often diluted the strategy, fragmenting the LLM's ability to commit to a coherent plan.

**Strategy and Discovery** HexMachina supports two complementary modes. In the *discovery phase*, it induces executable artifacts such as an `adapters.py` file that stabilizes access to the Catanatron API without any human documentation. This ensures that later improvements build on a reliable, reusable interface. In the *improvement phase*, the system searches over new tactics, revisits prior players, and integrates insights from past runs. Together, these phases ensure that learning is both grounded in the environment and continuously refined across evolutions.

**Orchestration** The orchestrator serves as the global planner, deciding when to analyze results, request new code, or revisit prior knowledge. Autonomy is enforced by a closed loop: the orchestrator makes high-level decisions based on game outcomes, artifacts, and agent communication, then delegates low-level tasks to the Analyzer and Coder. This separation prevents the system from stalling on details while still maintaining tight control over long-term strategy evolution.

**Memory** Finally, HexMachina maintains both *game memory* and *semantic memory* across evolutions. Game memory archives past players, their code, and evaluation artifacts, enabling direct comparisons and reuse of successful strategies. Semantic memory allows each agent to persist expertise relevant to its role, e.g., the Coder retaining knowledge of syntax patterns or the Analyst preserving diagnostic heuristics. This dual memory system underpins continual learning: instead of starting from scratch each evolution, the system accumulates strategic and technical knowledge that compounds over time.

## 4.2 AGENTS

Each agent in HexMachina is a specialist that can call tools (up to 5 per turn) and then yield a single, compact message back to the main loop. Only this final message is persisted to memory, ensuring concise, role-specific contributions. Each agent has access to a *Think Tool* (adapted from Langchain's deep research agent) to support internal reasoning and explainability. Inputs and outputs are standardized, enabling interchangeable models and providers.

Importantly, not all agents are equally useful in every phase. In the *discovery phase*, the full set of agents contributes to inducing a stable `adapters.py` file from scratch. However, in the *improvement phase*, we found that the most effective configuration is a streamlined loop of **Orchestrator, Analyst, and Coder**. Additional recommendations from the Strategist and Researcher often diluted coherence, so these roles are used only during discovery or when revisiting artifacts, not for direct strategy refinement.

**Orchestrator:** Global planner and orchestrator.
  *Inputs:* Orchestrator Messages and summary of evolution
  *Outputs:* Thoughts, system goal, next chosen agent, next agent objective
  *Tools:* None

**Coder:** Turn strategies into compilable code.
  *Inputs:* Objective from Orchestrator, adapter contents
  *Outputs:* Executable code, summary of changes
  *Tools:* Write/edit file

**Analyst:** Experimentation evaluator
  *Inputs:* Objective from Orchestrator, summary of evolution, current player and Coder summary of changes, game artifacts, adapter contents
  *Outputs:* Post-game diagnosis, specific analysis, adapter failure
  *Tools:* Read local file

**Researcher:** Recover API/engine facts and domain tactics. Primarily active during discovery.
  *Inputs:* Objective from Orchestrator, list of files, adapter contents
  *Outputs:* Citations, code pointers, or concise notes with source references
  *Tools:* Read local file, web search

**Strategist:** Propose concrete, testable plans. Primarily active during discovery.
  *Inputs:* Orchestrator Objective, Evolution Summary, current player, adapter contents
  *Outputs:* Strategy spec and evaluation
  *Tools:* Read local file, view older experiment, web search

## 5 EXPERIMENT SETUP

We evaluate HexMachina in the open-source *Catanatron* environment under controlled 2-player, 10-point Catan games. Each experiment consists of repeated head-to-head matches against the strongest built-in heuristic bot, *AlphaBeta*. We measure both *win rate* and *final victory points* as indicators of strategic quality. Games are deterministic given a random seed, allowing us to reproduce results and separate genuine improvements from stochastic variance. Data was collected over 60 hours across two machines (MacBook Pro 2019, 16GB; MacBook M1 Max 2021, 32GB).

### 5.1 BASELINES

Our baselines capture a spectrum of reference points, from trivial random play to a strong, hand-engineered heuristic, allowing us to contextualize HexMachina's performance against both naive policies and established rule-based expertise.

**Random.** The simplest control agent chooses uniformly from the legal action space each turn. While strategically meaningless, this baseline sets a lower bound for performance and highlights how much structure even a minimal policy adds.

**LLM Player.** We also evaluate a Reflexion-style agent (Shinn et al. (2023)) that reformats the game state into text and queries Claude 3.7 once per turn with a high-level goal. This baseline reflects the "prompt-centric" paradigm: the LLM directly drives play without any compiled memory or artifact reuse. Due to inference cost (approx. 70 queries per game), we limited this evaluation to 20 games with a model we had free access too, but it provides a critical comparison to show how quickly context saturation and lack of persistence hinder long-horizon play.

**Basic Continual Learner (HexMachina w/o discovery).** To isolate the value of separating discovery and improvement, we also test a single-phase continual learning setup equivalent to Hex-Machina without the discovery phase. Here the system attempts to learn both the environment interface and the strategy simultaneously. This resembles prior lifelong agents such as Voyager and Eureka (Wang et al. (2023a); Ma et al. (2024)), which evolve strategies directly from raw interaction. As shown later in Appendix A.2, these agents often converge on shallow heuristics (e.g., one-ply VP-only evaluators), highlighting the difficulty of strategic refinement without first stabilizing the interface.

**AlphaBeta.** Finally, we include Catanatron's AlphaBeta agent, a depth-2 minimax with stochastic expansion and heuristic evaluation. This player is fast, strong, and widely used as a benchmark; in self-play it achieves a 50% win rate by construction. It represents the ceiling for our experiments, providing a human-engineered reference against which HexMachina's evolved players can be meaningfully compared.

## 5.2 MODELS

HexMachina is model-agnostic, but in practice we deploy different LLMs for different roles to balance strength and efficiency. We test three orchestrator backends, GPT-5-mini, Claude 3.7, and Mistral-large, to assess robustness across providers. Unless otherwise noted, GPT-5-mini is used for the Coder, which requires reliable code synthesis, while Mistral-large is assigned to support roles (Analyst, Strategist, and Researcher) to reduce cost and latency. This division reflects a general principle of our framework: leverage stronger models where precision is critical (e.g., code generation) and more efficient models where interpretive or diagnostic reasoning suffices.

## 6 RESULTS AND DISCUSSION

### 6.1 CONTINUAL LEARNING

We first examine the impact of continual learning through evolution runs of 10 steps, with each step evaluating FooPlayer across 30 games. Figure 3 shows HexMachina steadily improving against AlphaBeta, eventually achieving parity and surpassing baseline players. A central design choice was the separation of *discovery* (API induction and artifact stabilization) from *improvement* (strategy evolution). Our experiments confirm that this separation is critical: systems without discovery struggled to stabilize player code, while those with discovery reliably produced executable players that improved across evolutions.

Interestingly, we found that HexMachina performed better when the Strategist and Researcher agents were *removed*, leaving only the Orchestrator, Analyst, and Coder. While the Strategist was intended to propose concrete plans, results suggest that LLMs often formulate effective strategies in a single shot, and passing these through multiple roles may dilute coherence. Thus, we report results using this streamlined configuration. This insight highlights a broader implication for continual learning: modular multi-agent systems are powerful, but not all roles contribute equally, and reducing mediation can strengthen strategic consistency.

Figure 4 provides a qualitative example of evolution in action. We observe HexMachina iteratively proposing, coding, and refining player strategies while preserving functional artifacts. This illustrates how artifact-centric continual learning transforms an LLM from a per-turn decision maker into a higher-level strategy designer with consistent policy execution.

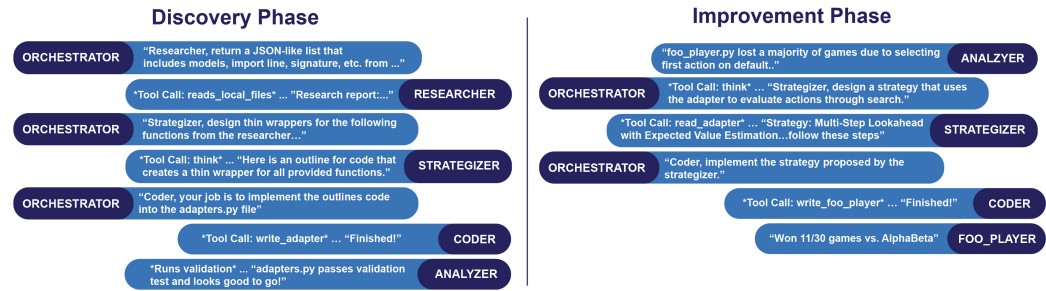

Figure 3: HexMachina Evolving to Outperform Existing Players

**Discovery Phase**

ORCHESTRATOR — "Researcher, return a JSON-like list that includes models, import line, signature, etc. from ..."

*Tool Call: reads_local_files* ... "Research report:..." — RESEARCHER

ORCHESTRATOR — "Strategizer, design thin wrappers for the following functions from the researcher…"

*Tool Call: think* ... "Here is an outline for code that creates a thin wrapper for all provided functions." — STRATEGIZER

ORCHESTRATOR — "Coder, your job is to implement the outlines code into the adapters.py file"

*Tool Call: write_adapter* ... "Finished!" — CODER

*Runs validation* ... "adapters.py passes validation test and looks good to go!" — ANALYZER

**Improvement Phase**

"foo_player.py lost a majority of games due to selecting first action on default.." — ANALZYER

ORCHESTRATOR — *Tool Call: think* ... "Strategizer, design a strategy that uses the adapter to evaluate actions through search."

*Tool Call: read_adapter* ... "Strategy: Multi-Step Lookahead with Expected Value Estimation…follow these steps" — STRATEGIZER

ORCHESTRATOR — "Coder, implement the strategy proposed by the strategizer."

*Tool Call: write_foo_player* ... "Finished!" — CODER

"Won 11/30 games vs. AlphaBeta" — FOO_PLAYER

Figure 4: Evolution Messages Example Dialogue

## 6.2 PLAYER COMPARISON

To stress test the best-evolved players, we ran each configuration 10 times with 100 games per run. Results are summarized in Table 2. HexMachina's best model (GPT-5-mini) reached a 54.1% win rate and $8.2 \pm 0.1$ victory points, matching or slightly exceeding AlphaBeta's 51.0% win rate and $7.8 \pm 0.2$ points. By contrast, the no-discovery baseline plateaued at much lower win rates, producing players that often failed to generalize beyond static heuristics.

A representative no-discovery agent (Appendix A.2) highlights why this baseline performs poorly. It carries out only a 1-ply lookahead, scoring states almost entirely on current victory points with trivial tie-breakers such as settlements, cities, or roads. With rollouts disabled and no modeling of stochastic production or opponent actions, it assigns identical scores to materially different choices, leading to random tie-breaking, poor settlement placement, and ineffective robber usage. These flaws explain its consistently weak performance in Table 2.

By contrast, the best evolved `FooPlayer` (Appendix A.1) demonstrates the benefits of HexMachina's discovery-improvement split. This agent combines phase-aware priorities (early expansion, mid-game balance, late-game upgrades), explicit heuristics for production diversity and robber disruption, and shallow rollouts that anticipate near-term outcomes. These capabilities yield stronger growth, better-timed upgrades, and consistent disruptive pressure on the opponent. The qualitative differences map directly onto the quantitative results in Table 2, underscoring that discovery is critical for stabilizing adapters and enabling the emergence of richer strategies.

Table 2: Win Rate and Victory Points for HexMachina compared to Baselines

| Player | Model | Win Rate | Victory Points |
|---|---|---|---|
| HexMachina | **GPT** | **54.1% [51%, 57%]** | **8.2 ± 0.1** |
| | Mistral | 49.2% [46%, 52%] | 7.8 ± 0.2 |
| | Claude | 38.4% [35%, 41%] | 7.2 ± 0.2 |
| LLM Player | Claude | 16.4% [3%, 30%] | 5.2 ± 1.2 |
| Alpha-Beta | X | 51.0% [48%, 54%] | 7.8 ± 0.2 |
| Random | X | 0.2% [0%, 0%] | 2.4 ± 0.0 |

## 6.3 ABLATIONS

To isolate the importance of individual design choices, we conducted ablation studies with three independent runs of 10 steps, each tested on 30 games. Results are shown in Table 3. Interestingly, removing the Strategist and Researcher improved performance relative to the full system, reaffirming our earlier finding that direct orchestration leads to clearer strategy translation. Removing the Analyst heavily impacted success as the agent is required to diagnose issues. There would often be situations where the system failed to recognize when functions were being mis-referenced from `adapters.py` without the Analyst bringing it into a failure loop. Overall, these findings back our

Table 3: Multi-Agent Architecture Ablations for HexMachina Policy Evolution

| Ablation | Win Rate | Victory Points |
|---|---|---|
| All Agents | 49.7% | 8.0 |
| No Analyst | 0.0% | 2.1 |
| **No Strategist + Researcher** | **54.1%** | **8.2** |

contribution statements: (1) HexMachina evolves executable strategies that rival top human-crafted baselines; (2) artifact preservation and doc-free discovery are essential to this success; (3) LLMs are best deployed at the level of strategy design, not per-turn play; and (4) multi-agent modularity is powerful, but optimal performance may emerge from leaner configurations that avoid unnecessary role handoffs.

## 7 CONCLUSION

Despite strong results, several limitations hindered performance from improving further. First, we evaluated players solely with win rate and final victory points, coarse metrics that sometimes mask subtler strengths and weaknesses. Second, the LLM occasionally hallucinated code or heuristics, requiring additional filtering. Third, the system was expensive to run due to inference costs, restricting the number of trials. Finally, performance remained closely tied to the quality of the underlying model, with more capable backends producing stronger players. Even with these constraints, HexMachina was able to autonomously induce an API, evolve a robust player, and achieve parity with AlphaBeta, the strongest human-crafted bot.

Looking forward, we see several avenues for advancement. Other researchers could attempt to design a more powerful multi-agent system on this benchmark or build a stronger hand-crafted player for comparison. More broadly, HexMachina should be tested on continual learning benchmarks beyond Catan to validate generality. Finally, the current 20-step evolution limit could be extended with improved memory and player management, enabling longer training horizons and more sophisticated strategies. Together, these extensions would push LLM agents closer to reliable long-horizon autonomy.

## 8 ETHICAL STATEMENT

Our system executes code in a closed loop with strict safeguards: generated programs run only within a controlled evaluation harness, preventing arbitrary system access. All experiments were logged with fixed random seeds and configuration files, ensuring transparency and reproducibility. While we present HexMachina as an autonomous agent, we avoid anthropomorphizingâĂŤour system is a code-evolving tool, not a sentient entity.

## 9 REPRODUCIBILITY STATEMENT

We release all code, experiment harnesses, and configuration files alongside this submission. To reproduce our results, clone the repository, install dependencies, and follow the step-by-step README instructions. Running experiments requires API keys for the tested LLMs; once provided, the system can be executed exactly as described to replicate all tables and figures.

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

# A APPENDIX

## A.1 HEXMACHINA'S BEST STRATEGY

```python
import random
from typing import Iterable, List, Optional, Any, Tuple

# MUST use the adapters surface to interact with the game environment
from .adapters import (
```

```
648        Game,
649        Player,
650        Color,
651        copy_game,
652        execute_deterministic,
653        execute_spectrum,
654        expand_spectrum,
655        list_prunned_actions,
656        prune_robber_actions,
657        base_fn,
658        value_production,
659        get_value_fn,
660    )
661
662    class FooPlayer(Player):
663        """A Foo player with game-phase aware decisioning, improved sampling,
664        short rollouts, and richer heuristics.
665
666        This implementation is defensive: it uses only the adapters surface
667                                            and
668        contains many fallbacks when attributes or adapter helpers are
669                                            missing.
670
671        Key features:
672        - Game-phase detection (early/mid/late) to bias settlement/road vs
673                                            city/dev-card
674        - Settlement & road potential heuristics to encourage early expansion
675        - Robber/knight evaluation to value disruption and steals
676        - Must-include guarantees for critical action types (settlement/road/
677                                            robber/dev)
678        - Rollout policy biased by phase and includes a light opponent-
679                                            response
680
681        NOTE: Many game model attribute names vary across environments. This
682                                            code
683        attempts multiple common attribute names and falls back to string-
684                                            based
685        heuristics when necessary. If the next run raises AttributeError for
686                                            an
687        adapters function or a specific attribute, provide the traceback so
688                                            it can
689        be patched to the concrete environment.
690        """
691
692        # Tunable constants (exposed to edit for experimentation)
693        MAX_SIMULATIONS = 24
694        PREFILTER_TOP_K = 8
695        ROLLOUT_DEPTH = 2
696        SIMULATION_BUDGET = 60
697        DEBUG = False
698
699        # Phase thresholds (used by get_game_phase)
700        EARLY_TURN_THRESHOLD = 20
701        MID_TURN_THRESHOLD = 45

        # Phase multipliers matrix (explicit)
        MULTS = {
            "EARLY": {"settlement": 2.0, "road": 1.8, "city": 0.8, "dev": 1.2
                                            },
            "MID": {"settlement": 1.0, "road": 1.0, "city": 1.25, "dev": 1.0}
                                            ,
            "LATE": {"settlement": 0.8, "road": 0.9, "city": 1.5, "dev": 1.0}
                                            ,
        }
```

```
702
703     # Must-include action tokens (robust, lowercase matching)
704     MUST_INCLUDE_TOKENS = {
705         "build_city",
706         "build_settlement",
707         "build_sett",
708         "build_road",
709         "buy_dev",
710         "buy_dev_card",
711         "buycard",
712         "play_knight",
713         "knight",
714         "move_robber",
715         "move_robber_action",
716         "robber",
717         "trade",
718         "offer_trade",
719     }
720
721     # Robber scoring base (increased)
722     ROBBER_BASE_SCORE = 80.0
723     ROBBER_BASE_SCORE_HIGH = 80.0
724
725     # Settlement target in early game
726     TARGET_SETTLEMENTS_EARLY = 3
727
728     # Epsilon-greedy randomness to avoid predictability
729     EPSILON_GREEDY = 0.04
730
731     # Rollout bonuses for the very first rollout step
732     ROLLOUT_SETTLEMENT_BONUS = 1.7
733     ROLLOUT_ROAD_BONUS = 1.4
734
735     # Tie tolerance
736     TOLERANCE = 1e-6
737
738     # Development card deck & EV constants
739     DEV_DECK = {"knight": 14, "vp": 5, "road_building": 2, "
740                                  year_of_plenty": 2, "monopoly":
741                                  2}
742     DEV_TOTAL = sum(DEV_DECK.values())
743     EV_KNIGHT = 0.15
744     EV_VP = 1.0
745     EV_ROAD_BUILDING = 0.25
746     EV_YOP = 0.2
747     EV_MONOPOLY = 0.3
748     DEV_EV_SCALE = 60.0
749     DEV_EV_THRESHOLD = 0.25
750
751     # Knight bonuses
752     KNIGHT_LARGEST_ARMY_BONUS = 50.0
753     KNIGHT_BASE = 25.0
754     KNIGHT_MIN_SCORE = 35.0
755
        # City/road/robber tuning (from latest analyzer guidance)
        CITY_URGENCY_BONUS = 85.0
        CITY_AFFORD_STRICT_ORE = 3
        CITY_AFFORD_STRICT_WHEAT = 2
        CITY_AFFORD_SOON_ORE = 2
        CITY_AFFORD_SOON_WHEAT = 1
        ROLLOUT_CITY_BONUS = 1.8
        ROAD_SCORE_BOOST = 9.0
        PROD_LOSS_IMPORTANCE = 70.0
        HIGH_VALUE_RESOURCE_SET = {"ore","wheat","metal","grain"}
        CITY_TIE_EPS = 0.02
```

```python
        # Forcing behavior flags and diagnostic counters
        PREFILTER_FORCE_CITY_IF = True
        CITY_FORCE_AFFORD_STRICT = True
        DEBUG_COUNTS = False

        def __init__(self, name: Optional[str] = None):
            super().__init__(Color.BLUE, name)
            # Try to cache a base value function from adapters
            try:
                self._value_fn = base_fn()
                self.debug_print("FooPlayer: Using adapters.base_fn() for
                                                    evaluation")
            except Exception as e:
                self._value_fn = None
                self.debug_print("FooPlayer: adapters.base_fn() not available
                                                    , will use heuristic.
                                                    Error:", e)

            # Diagnostic counters (quiet unless DEBUG)
            self._diag_forced_settlement = 0
            self._diag_forced_road = 0
            self._diag_city_urgency_count = 0
            self._diag_settle_urgency_count = 0

            # New counters for tuning
            self.COUNTER_FORCED_CITY = 0
            self.COUNTER_DEV_BUY_FORCED = 0
            self.COUNTER_BUY_DEV_ACTUALLY = 0
            self.COUNTER_BUILD_CITY_ACTUALLY = 0
            self.COUNTER_ROBBER_ACTUALLY = 0

    # ------------------- Debug helper -------------------
    def debug_print(self, *args: Any) -> None:
        if self.DEBUG:
            print(*args)

    # ------------------- Utility helpers -------------------
    def _get_player_color(self) -> Color:
        """Return this player's color. Try common attribute names."""
        if hasattr(self, "color"):
            return getattr(self, "color")
        if hasattr(self, "_color"):
            return getattr(self, "_color")
        return Color.BLUE

    def _safe_action_name(self, action: Any) -> str:
        """Produce a lowercase string name for the action for robust
                                        matching."""
        try:
            at = getattr(action, "action_type", None)
            if at is None:
                at = getattr(action, "type", None)
            if at is not None:
                try:
                    return str(at.name).lower()
                except Exception:
                    return str(at).lower()
        except Exception:
            pass
        try:
            # Some Action objects have a .name or .action_name
            name = getattr(action, "name", None) or getattr(action, "
                                        action_name", None)
            if name is not None:
```

```
810                    return str(name).lower()
811            except Exception:
812                pass
813            try:
814                return str(action).lower()
815            except Exception:
816                return ""

817        # ------------------ Phase detection -------------------
818        def get_game_phase(self, game: Game, color: Optional[Color] = None) -
819                                        > str:
820            """Return 'EARLY', 'MID', or 'LATE' based on turn counters or VP
821                                        thresholds.

822            Order of checks:
823            1) turn/tick counters if available (preferred)
824            2) max VP among players
825            3) fallback to conservative MID
826            """
827            try:
828                state = getattr(game, "state", game)
829                turn_count = (
830                    getattr(state, "turn", None)
831                    or getattr(state, "tick", None)
832                    or getattr(state, "turn_count", None)
833                    or getattr(state, "tick_count", None)
834                )
835                if isinstance(turn_count, (int, float)):
836                    tc = int(turn_count)
837                    if tc < self.EARLY_TURN_THRESHOLD:
838                        return "EARLY"
839                    if tc < self.MID_TURN_THRESHOLD:
840                        return "MID"
841                    return "LATE"
842            except Exception:
843                pass

844            # Fallback: use maximum VP among players
845            try:
846                state = getattr(game, "state", game)
847                players = getattr(state, "players", None) or getattr(game, "
848                                            players", None) or []
849                max_vp = 0
850                if isinstance(players, dict):
851                    for p in players.values():
852                        vp = getattr(p, "victory_points", None) or getattr(p,
853                                                "vp", None) or
854                                                0
855                        try:
856                            vp = int(vp)
857                        except Exception:
858                            vp = 0
859                        max_vp = max(max_vp, vp)
860                else:
861                    for p in players:
862                        vp = getattr(p, "victory_points", None) or getattr(p,
863                                                "vp", None) or
                                                0
                        try:
                            vp = int(vp)
                        except Exception:
                            vp = 0
                        max_vp = max(max_vp, vp)
                if max_vp < 4:
                    return "EARLY"
```

```
864             if max_vp < 8:
865                 return "MID"
866             return "LATE"
867         except Exception:
868             # Conservative fallback to MID
869             return "MID"
870
871     # ------------------ Heuristic / evaluation (phase-aware)
                                           --------------------
872     def _heuristic_value(self, game: Game, color: Color) -> float:
873         """Phase-aware heuristic including production potential and city-
                                            upgrade progress.
874
875         Many attribute names are attempted to be robust across different
876                                           game models.
877         """
878         # Die probabilities for numbers 2..12 ignoring 7
879         die_prob = {2: 1 / 36, 3: 2 / 36, 4: 3 / 36, 5: 4 / 36, 6: 5 / 36
880                                         , 8: 5 / 36, 9: 4 / 36, 10:
881                                         3 / 36, 11: 2 / 36, 12: 1 /
882                                         36}
883         # Player lookup
884         player_state = None
885         try:
886             state = getattr(game, "state", game)
887             players = getattr(state, "players", None) or getattr(game, "
                                            players", None)
888             if isinstance(players, dict):
889                 player_state = players.get(color) or players.get(str(
                                            color))
890             elif isinstance(players, (list, tuple)):
891                 for p in players:
892                     if getattr(p, "color", None) == color or getattr(p, "
                                                color", None) ==
                                                str(color):
894                         player_state = p
895                         break
896         except Exception:
897             player_state = None
898
899         def _safe_get(obj, *names, default=0):
900             if obj is None:
901                 return default
902             for name in names:
903                 try:
904                     val = getattr(obj, name)
905                     if val is not None:
906                         return val
907                 except Exception:
908                     try:
909                         val = obj[name]
910                         if val is not None:
911                             return val
912                     except Exception:
913                         continue
914             return default
915
916         vp = _safe_get(player_state, "victory_points", "vp", default=0)
917         settlements = _safe_get(player_state, "settlements", "
                                        settle_count", "
                                        settle_locations", default=0
                                        )
        if isinstance(settlements, (list, tuple)):
            settlements = len(settlements)
```

```
918          cities = _safe_get(player_state, "cities", "city_count", "
919                                             city_locations", default=0)
920          if isinstance(cities, (list, tuple)):
921              cities = len(cities)
922          roads = _safe_get(player_state, "roads", "road_count", default=0)
923          if isinstance(roads, (list, tuple)):
924              roads = len(roads)
925          dev_vp = _safe_get(player_state, "dev_vp", "dev_victory_points",
                                             default=0)
926
927          # Resources summary
928          resources_obj = _safe_get(player_state, "resources", default=0)
929          resources_total = 0
930          resource_diversity = 0
931          try:
932              if isinstance(resources_obj, dict):
933                  resources_total = sum(resources_obj.values())
934                  resource_diversity = sum(1 for v in resources_obj.values
                                                 () if v > 0)
935              elif isinstance(resources_obj, (list, tuple)):
936                  resources_total = sum(resources_obj)
937                  resource_diversity = sum(1 for v in resources_obj if v >
                                                 0)
938              else:
939                  resources_total = int(resources_obj)
940                  resource_diversity = 1 if resources_total > 0 else 0
941          except Exception:
942              resources_total = 0
943              resource_diversity = 0
944
945          # Production potential estimation
946          prod_value = 0.0
947          try:
948              board = getattr(state, "board", None) or getattr(game, "board
                                             ", None)
949              hexes = getattr(board, "hexes", None) or getattr(board, "
                                             tiles", None) or []
950              settlements_list = _safe_get(player_state, "settlements", "
                                             settle_locations",
                                             default=[])
951              if isinstance(settlements_list, (list, tuple)):
952                  for s in settlements_list:
953                      try:
954                          for h in hexes:
955                              neighbors = getattr(h, "vertices",
                                             None) or getattr(h, "adjacent_vertices",
                                             None) or []
956                              if s in neighbors:
957                                  num = getattr(h, "roll",
                                             None) or getattr(h, "number",
                                             None) or getattr(h, "value",
                                             None)
958                                  try:
959                                      num = int(num)
960                                  except Exception:
961                                      num = None
962                                  if num in die_prob:
963                                      prod_value += die_prob[num] * 1.0
964                      except Exception:
965                          continue
966              cities_list = _safe_get(player_state, "cities", "
                                             city_locations", default
                                             =[])
967              if isinstance(cities_list, (list, tuple)):
968                  for c in cities_list:
```

```python
                        try:
                            for h in hexes:
                                neighbors = getattr(h, "vertices",
                                None) or
                                getattr(h, "adjacent_vertices", None) or []
                                if c in neighbors:
                                    num = getattr(h, "roll",
                                    None) or getattr(h, "number",
                                    None) or getattr(h, "value",
                                    None)
                                    try:
                                        num = int(num)
                                    except Exception:
                                        num = None
                                    if num in die_prob:
                                        prod_value += die_prob[num] * 2.0
                        except Exception:
                            continue
            except Exception:
                prod_value = 0.0

        # City upgrade progress heuristic
        city_resource_val = 0.0
        try:
            if isinstance(resources_obj, dict):
                wheat = resources_obj.get("wheat", 0) + resources_obj.get
                                                        ("grain", 0)
                ore = resources_obj.get("ore", 0) + resources_obj.get("
                                                        metal", 0)
                city_resource_val = min(wheat, ore)
        except Exception:
            city_resource_val = 0.0

        # Phase multipliers
        phase = self.get_game_phase(game, color)
        mults = self.MULTS.get(phase, self.MULTS["MID"])
        settlement_mul = mults["settlement"]
        road_mul = mults["road"]
        city_mul = mults["city"]
        dev_mul = mults["dev"]

        # Adjust production weight by phase
        prod_weight = 80.0 if phase == "EARLY" else 45.0 if phase == "MID
                                            " else 30.0

        # Compose weighted sum (city reward scaled by city_mul)
        score = (
            float(vp) * 100.0
            + float(settlements) * 25.0 * settlement_mul
            + float(cities) * 60.0 * city_mul
            + float(roads) * 6.0 * road_mul
            + float(dev_vp) * 50.0
            + float(resources_total) * 1.0
            + float(resource_diversity) * 3.0
            + float(city_resource_val) * 5.0
            + float(prod_value) * prod_weight
        )

        return float(score)

    def _evaluate_game_state(self, game: Game, color: Color) -> float:
        """Evaluate a single game state for the given player color.

        Prefer adapters.base_fn() if available (cached in self._value_fn)
                                            . If available, combine
```

```
            it with the heuristic for stability. We keep phase multipliers
                                          inside the heuristic so
            they influence the final blended value.
            """
            heuristic = self._heuristic_value(game, color)
            if self._value_fn is not None:
                try:
                    vf_val = float(self._value_fn(game, color))
                    return 0.85 * vf_val + 0.15 * heuristic
                except Exception as e:
                    self.debug_print("FooPlayer: value_fn failed during
                                                  evaluate_game_state,
                                                   falling back to
                                                  heuristic. Error:",
                                                  e)
            return float(heuristic)

    # ------------------- Cheap scoring & potentials -------------------
    def _get_player_state(self, game: Game, color: Color) -> Any:
        """Return the player_state object from the game state (best-
                                          effort)."""
        try:
            state = getattr(game, "state", game)
            players = getattr(state, "players", None) or getattr(game, "
                                              players", None)
            if isinstance(players, dict):
                return players.get(color) or players.get(str(color))
            elif isinstance(players, (list, tuple)):
                for p in players:
                    if getattr(p, "color", None) == color or getattr(p, "
                                                  color", None) ==
                                                   str(color):
                        return p
        except Exception:
            return None
        return None

    def settlement_potential(self, action: Any, game: Game, color: Color)
                                      -> float:
        """Estimate benefit of a settlement action: new resource types
                                          and production.

        Best-effort: try to parse adjacent hexes from action or fallback
                                          to string heuristics.
        """
        bonus = 0.0
        try:
            name = self._safe_action_name(action)
            # Quick check: if action indicates a settlement, give base
            if any(tok in name for tok in ("build_settlement", "
                                          build_sett", "settle")):
                bonus += 5.0

            # Try to parse a vertex index from the action string
            digits = [int(tok) for tok in name.split() if tok.isdigit()]
            vertex = digits[0] if digits else None

            state = getattr(game, "state", game)
            board = getattr(state, "board", None) or getattr(game, "board
                                              ", None)
            hexes = getattr(board, "hexes", None) or getattr(board, "
                                              tiles", None) or []

            # Player's current resource types
            player_state = self._get_player_state(game, color)
```

```
1080            player_types = set()
1081            try:
1082                settlements_list = getattr(player_state, "settlements",
1083                                                None) or getattr(
1084                                                player_state, "
1085                                                settle_locations",
1086                                                None) or []
1087                if isinstance(settlements_list, (list, tuple)):
1088                    for s in settlements_list:
1089                        for h in hexes:
1090                            neighbors = getattr(h, "vertices",
1091                            None) or getattr(h, "adjacent_vertices",
1092                            None) or []
1093                            if s in neighbors:
1094                                rtype = getattr(h, "resource",
1095                                None) or getattr(h, "type",
1096                                None)
1097                                if rtype is not None:
1098                                    player_types.add(str(rtype).lower())
1099            except Exception:
1100                player_types = set()
1101
1102            # Adjacent resources for proposed vertex
1103            adj_resources = set()
1104            prod_sum = 0.0
1105            die_prob = {2: 1 / 36, 3: 2 / 36, 4: 3 / 36, 5: 4 / 36, 6: 5
1106                                                / 36, 8: 5 / 36, 9: 4 /
1107                                                36, 10: 3 / 36, 11: 2 /
1108                                                36, 12: 1 / 36}
1109            if vertex is not None:
1110                for h in hexes:
1111                    try:
1112                        neighbors = getattr(h, "vertices",
1113                        None) or getattr(h, "adjacent_vertices",
1114                        None) or []
1115                        if vertex in neighbors:
1116                            rtype = getattr(h, "resource",
1117                            None) or getattr(h, "type",
1118                            None)
1119                            if rtype is not None:
1120                                adj_resources.add(str(rtype).lower())
1121                            num = getattr(h, "roll",
1122                            None) or getattr(h, "number",
1123                            None) or getattr(h, "value",
1124                            None)
1125                            try:
1126                                num = int(num)
1127                            except Exception:
1128                                num = None
1129                            if num in die_prob:
1130                                prod_sum += die_prob[num]
1131                    except Exception:
1132                        continue
1133            # New types
            new_types = adj_resources - player_types
            bonus += float(len(new_types)) * 12.0
            bonus += float(prod_sum) * 8.0
        except Exception:
            pass
        return float(bonus)

    def road_connection_potential(self, action: Any, game: Game, color:
                                        Color) -> float:
        """Estimate if a road action helps expansion. Best-effort using
                                        indices."""
```

```
1134            bonus = 0.0
1135            try:
1136                name = self._safe_action_name(action)
1137                # try to extract numbers from action name
1138                digits = [int(tok) for tok in name.split() if tok.isdigit()]
1139                # player's settlement/city vertices
1140                player_state = self._get_player_state(game, color)
1141                player_nodes = set()
1142                try:
1143                    settles = getattr(player_state, "settlements", None) or
                                            getattr(player_state
                                            , "settle_locations"
                                            , None) or []
1145                    cities = getattr(player_state, "cities", None) or getattr
                                            (player_state, "
                                            city_locations",
                                            None) or []
1149                    if isinstance(settles, (list, tuple)):
1150                        player_nodes.update(settles)
1151                    if isinstance(cities, (list, tuple)):
                            player_nodes.update(cities)
1152                except Exception:
1153                    player_nodes = set()
1154
1155                if digits:
                        # if any digit matches a player node, give higher bonus
1156                    if any(d in player_nodes for d in digits):
1157                        bonus += 6.0
1158                    else:
1159                        bonus += 3.0
1160                else:
                        # fallback string heuristics
1161                    if "build_road" in name or ("road" in name and "build" in
                                                name):
1163                        bonus += 2.0
1164            except Exception:
1165                pass
1166            return float(bonus)
1167
1168        def evaluate_buy_dev_card(self, action: Any, game: Game, color: Color
                                            ) -> bool:
1169            """Decide whether buying a dev card is currently a good idea (
                                            best-effort)."""
1170            try:
1171                player_state = self._get_player_state(game, color)
1172                resources = getattr(player_state, "resources", None)
1173                if isinstance(resources, dict):
1174                    ore = resources.get("ore", 0) + resources.get("metal", 0)
1175                    wheat = resources.get("wheat", 0) + resources.get("grain"
                                            , 0)
1176                    others = sum(v for k, v in resources.items() if k not in
                                            ("ore", "metal", "
                                            wheat", "grain"))
1179                    # if have ore+wheat+another, prefer dev card; or if no
                                            settlement/road/city
                                            affordable
1181                    if ore >= 1 and wheat >= 1 and others >= 1:
1182                        return True
1183                    # fallback: if early game and we have some resources but
                                            no settlement
                                            potential, allow dev
                                            buy
1186                    phase = self.get_game_phase(game, color)
1187                    if phase == "EARLY" and (ore + wheat + others) >= 3:
                            return True
```

```
1188            except Exception:
1189                pass
1190            return False
1191
1192        def dev_card_ev_estimate(self, game: Game, color: Color) -> float:
1193            """Estimate expected VP-equivalent value of buying a development
                                                card.
1194
1195            Uses static DEV_DECK and EV_* constants and scales by opponent
                                                pressure and army gaps.
1196
1197            Returns a small VP-equivalent number (e.g., ~0.3-0.6 when
                                                favorable).
1198            """
1199            try:
1200                base_ev = 0.0
1201                # composition-based base EV
1202                base_ev += (self.DEV_DECK.get("knight", 0) / self.DEV_TOTAL)
                                                * self.EV_KNIGHT
1203                base_ev += (self.DEV_DECK.get("vp", 0) / self.DEV_TOTAL) *
                                                self.EV_VP
1204                base_ev += (self.DEV_DECK.get("road_building", 0) / self.
                                                DEV_TOTAL) * self.
                                                EV_ROAD_BUILDING
1205                base_ev += (self.DEV_DECK.get("year_of_plenty", 0) / self.
                                                DEV_TOTAL) * self.EV_YOP
1206                base_ev += (self.DEV_DECK.get("monopoly", 0) / self.DEV_TOTAL
                                                ) * self.EV_MONOPOLY
1207
1208                # Scale factors: opponents production pressure and army
                                                proximity
1209                # Compute opponents' max production (best-effort)
1210                state = getattr(game, "state", game)
1211                board = getattr(state, "board", None) or getattr(game, "board
                                                ", None)
1212                hexes = getattr(board, "hexes", None) or getattr(board, "
                                                tiles", None) or []
1213
1214                opponents = []
1215                players = getattr(state, "players", None) or getattr(game, "
                                                players", None) or []
1216                my_color = color
1217                if isinstance(players, dict):
1218                    for k, p in players.items():
1219                        if k == my_color or getattr(p, "color", None) ==
                                                    my_color:
1220                            continue
1221                        opponents.append(p)
1222                else:
1223                    for p in players:
1224                        if getattr(p, "color", None) == my_color:
1225                            continue
1226                        opponents.append(p)
1227
1228                # compute simple production score for each opponent
1229                die_prob = {2: 1 / 36, 3: 2 / 36, 4: 3 / 36, 5: 4 / 36, 6: 5
                                                / 36, 8: 5 / 36, 9: 4 /
                                                36, 10: 3 / 36, 11: 2 /
                                                36, 12: 1 / 36}
1230                max_opp_prod = 0.0
1231                for opp in opponents:
1232                    prod = 0.0
1233                    opp_settles = getattr(opp, "settlements", None) or
                                                    getattr(opp, "
                                                    settle_locations",
                                                    None) or []
```

```
1242              opp_cities = getattr(opp, "cities", None) or getattr(opp,
1243                                                          "city_locations",
1244                                                          None) or []
1245          try:
1246              for s in opp_settles:
1247                  for h in hexes:
1248                      neighbors = getattr(h, "vertices",
1249                      None) or getattr(h, "adjacent_vertices",
1250                      None) or []
1251                      if s in neighbors:
1252                          num = getattr(h, "roll",
1253                          None) or getattr(h, "number",
1254                          None) or getattr(h, "value",
1255                          None)
1256                          try:
1257                              num = int(num)
1258                          except Exception:
1259                              num = None
1260                          if num in die_prob:
1261                              prod += die_prob[num]
1262              for c in opp_cities:
1263                  for h in hexes:
1264                      neighbors = getattr(h, "vertices",
1265                      None) or getattr(h, "adjacent_vertices",
1266                      None) or []
1267                      if c in neighbors:
1268                          num = getattr(h, "roll",
1269                          None) or getattr(h, "number",
1270                          None) or getattr(h, "value",
1271                          None)
1272                          try:
1273                              num = int(num)
1274                          except Exception:
1275                              num = None
1276                          if num in die_prob:
1277                              prod += 2.0 * die_prob[num]
1278          except Exception:
1279              pass
1280          max_opp_prod = max(max_opp_prod, prod)

        # army gap factor
        my_state = self._get_player_state(game, color)
        my_army = getattr(my_state, "army", None) or getattr(my_state
                                          , "army_size", None) or
                                          getattr(my_state, "
                                          knights_played", None)
                                          or 0
        try:
            my_army = int(my_army)
        except Exception:
            my_army = 0
        max_other_army = 0
        try:
            if isinstance(players, dict):
                for k, p in players.items():
                    if k == my_color or getattr(p, "color", None) ==
                                                        my_color:
                        continue
                    oa = getattr(p, "army", None) or
                    getattr(p, "army_size", None) or
                    getattr(p, "knights_played",
                    None) or 0
                    try:
                        oa = int(oa)
                    except Exception:
```

```
1296                             oa = 0
1297                         max_other_army = max(max_other_army, oa)
1298                 else:
1299                     for p in players:
1300                         if getattr(p, "color", None) == my_color:
1301                             continue
1302                         oa = getattr(p, "army", None) or
1303                         getattr(p, "army_size", None) or
1304                         getattr(p, "knights_played", None) or 0
1305                         try:
1306                             oa = int(oa)
1307                         except Exception:
1308                             oa = 0
1309                         max_other_army = max(max_other_army, oa)
1310         except Exception:
1311             max_other_army = 0
1312
1313         army_gap = max(0, max_other_army - my_army)
1314
1315         # scale base_ev conservatively
1316         scale = 1.0
1317         if max_opp_prod > 0.25:  # opponent has strong production
1318             scale += 0.25
1319         if army_gap >= 1:
1320             scale += 0.15 * army_gap
1321
1322         final_ev = base_ev * scale
1323         return float(final_ev)
1324     except Exception:
1325         # fallback conservative
1326         return 0.25
1327
1328 def build_urgency(self, game: Game, color: Color) -> Tuple[float,
1329                                     float, float]:
1330     """Return (city_bonus, settlement_bonus, road_bonus) depending on
1331                                     resources and phase."""
1332     city_bonus = 0.0
1333     settlement_bonus = 0.0
1334     road_bonus = 0.0
1335     try:
1336         player_state = self._get_player_state(game, color)
1337         resources = getattr(player_state, "resources", None) or {}
1338         if not isinstance(resources, dict):
1339             # try to coerce
1340             try:
1341                 total = sum(resources)
1342                 resources = {"res": total}
1343             except Exception:
1344                 resources = {}
1345
1346         # simple can_afford_city_soon heuristic
1347         ore = resources.get("ore", 0) + resources.get("metal", 0)
1348         wheat = resources.get("wheat", 0) + resources.get("grain", 0)
1349         settlements_list = getattr(player_state, "settlements", None)
                                     or getattr(player_state
                                     , "settle_locations",
                                     None) or []
         settlements_owned = len(settlements_list) if isinstance(
                                     settlements_list, (list,
                                     tuple)) else 0

         phase = self.get_game_phase(game, color)
         # If mid/late and can afford city soon, large city urgency
         if phase in ("MID", "LATE") and ore >= 2 and wheat >= 1:
             city_bonus += 40.0
```

```
                    self._diag_city_urgency_count += 1
            # If early and lacking settlements target, encourage
                                           settlements strongly
            if phase == "EARLY" and settlements_owned < self.
                                      TARGET_SETTLEMENTS_EARLY
                                      :
                settlement_bonus += 35.0
                self._diag_settle_urgency_count += 1
            # Road potential: give moderate constant bonus
            road_bonus += 10.0
        except Exception:
            pass
        return city_bonus, settlement_bonus, road_bonus

    def cheap_pre_score(self, action: Any, game: Game, color: Color) ->
                                     float:
        """Cheap, fast scoring used to prioritize actions for simulation
                                     (phase-aware)."""
        s = 0.0
        name = self._safe_action_name(action)

        phase = self.get_game_phase(game, color)
        mults = self.MULTS.get(phase, self.MULTS["MID"])
        settlement_mul = mults["settlement"]
        road_mul = mults["road"]
        city_mul = mults["city"]
        dev_mul = mults["dev"]

        # urgency bonuses
        city_urgency, sett_urgency, road_urgency = self.build_urgency(
                                     game, color)

        # Reward direct VP gains but adjust city bias early
        if any(tok in name for tok in ("build_city",)):
            base_city = max(50.0, 100.0 * city_mul - 15.0)
            # penalize city if early and still below settlement target
            try:
                player_state = self._get_player_state(game, color)
                settles = getattr(player_state, "settlements", None) or
                                         getattr(player_state
                                         , "settle_locations"
                                         , None) or []
                curr_settlements = len(settles) if isinstance(settles, (
                                         list, tuple)) else 0
                if phase == "EARLY" and curr_settlements <
                self.TARGET_SETTLEMENTS_EARLY:
                    base_city *= 0.6
            except Exception:
                pass
            s += base_city + city_urgency

        if any(tok in name for tok in ("build_settlement", "build_sett"))
                                     :
            s += 90.0 * settlement_mul
            # add settlement potential (resource diversity / production)
            s += self.settlement_potential(action, game, color) * (1.0 if
                                          phase != "EARLY" else
                                         settlement_mul)
            s += sett_urgency

        if "buy_dev" in name or "buycard" in name or "buy_dev_card" in
                                         name:
            # compute EV estimate
            dev_ev = self.dev_card_ev_estimate(game, color)
            s += dev_ev * self.DEV_EV_SCALE
```

```
1404                    # slightly reduced base bias to favor cities when urgent
1405                    if self.evaluate_buy_dev_card(action, game, color):
1406                        s += 8.0 * dev_mul
1407                    try:
1408                        if dev_ev >= self.DEV_EV_THRESHOLD:
1409                            s += 2.0
1410                    except Exception:
1411                        pass
1412
1413                if "build_road" in name or ("road" in name and "build" in name):
1414                    s += 20.0 * road_mul
1415                    s += self.road_connection_potential(action, game, color) * (1
                                                          .0 if phase != "EARLY"
1416                                                          else road_mul)
1417                    s += road_urgency
1418
1419                if "knight" in name or "play_knight" in name:
1420                    # raise baseline and include army/steal bonuses
1421                    s += 70.0
1422                    s += self.evaluate_play_knight(action, game, color)
1423
1424                if "robber" in name or "move_robber" in name:
1425                    s += 50.0
1426                    s += self.evaluate_robber_action(action, game, color)
1427
1428                if "trade" in name or "offer_trade" in name:
1429                    s += 10.0
1430
1431                # Encourage hitting settlement target early
1432                try:
1433                    player_state = self._get_player_state(game, color)
1434                    curr_settlements = 0
1435                    settles = getattr(player_state, "settlements", None) or
                                                      getattr(player_state, "
1436                                                      settle_locations", None)
                                                       or []
1437                    if isinstance(settles, (list, tuple)):
1438                        curr_settlements = len(settles)
1439                    if phase == "EARLY" and curr_settlements < self.
1440                                                      TARGET_SETTLEMENTS_EARLY
1441                                                       and any(tok in name for
1442                                                      tok in ("
1443                                                      build_settlement", "
1444                                                      build_sett")):
1445                        s += 30.0
1446                except Exception:
1447                    pass
1448
1449                # small settlement/road potentials for other actions
1450                if not any(tok in name for tok in ("build_settlement", "
1451                                                  build_sett")):
1452                    s += self.settlement_potential(action, game, color) * 0.1
1453                if not any(tok in name for tok in ("build_road",)):
1454                    s += self.road_connection_potential(action, game, color) * 0.
                                                      1

                # Minor random tie-break
                s += random.random() * 1e-3
                return s

    # ------------------ Prefilter actions (phase-aware guarantees)
                                          ------------------
    def prefilter_actions(self, actions: List[Any], game: Game, color:
                                          Color) -> List[Any]:
```

```
1458          """Return a bounded list of candidate actions to evaluate
1459                                          thoroughly.
1460
1461          Guarantees inclusion of must-include tokens and early-game
1462                                          settlement/road actions.
1463          """
1464          if not actions:
1465              return []
1466
1467          all_actions = list(actions)
1468          phase = self.get_game_phase(game, color)
1469
1470          musts = []
1471          others = []
1472          found_settlement = None
1473          found_road = None
1474          for a in all_actions:
1475              name = self._safe_action_name(a)
1476              if any(tok in name for tok in self.MUST_INCLUDE_TOKENS):
1477                  if a not in musts:
1478                      musts.append(a)
1479              else:
1480                  others.append(a)
1481              if found_settlement is None and any(tok in name for tok in ("
1482                                                  build_settlement", "
1483                                                  build_sett", "settle")):
1484                  found_settlement = a
1485              if found_road is None and any(tok in name for tok in ("
1486                                                  build_road", "road")):
1487                  found_road = a

         # Phase-based forced includes: ensure at least one settlement and
                                          one road action if present
                                          in EARLY
         if phase == "EARLY":
             if found_settlement is not None and found_settlement not in
                                                  musts:
                 musts.append(found_settlement)
                 self._diag_forced_settlement += 1
             if found_road is not None and found_road not in musts:
                 musts.append(found_road)
                 self._diag_forced_road += 1

         # Include recommended dev-card buys if conservative and EV
                                          threshold met
         for a in all_actions:
             name = self._safe_action_name(a)
             if any(tok in name for tok in ("buy_dev", "buycard", "
                                                  buy_dev_card")):
                 try:
                     if self.evaluate_buy_dev_card(a, game, color):
                         dev_ev = self.dev_card_ev_estimate(game, color)
                         if dev_ev >= self.DEV_EV_THRESHOLD and a not in
                                                              musts:
                             # include only if dev EV merits it
                             musts.append(a)
                 except Exception:
                     pass

         # Ensure robber/knight actions are present
         for a in all_actions:
             name = self._safe_action_name(a)
             if any(tok in name for tok in ("robber", "move_robber", "
                                                  knight", "play_knight"))
                                                  :
```

```
1512                    if a not in musts:
1513                        musts.append(a)
1514
1515            # Score and pick top-K from others
1516            scored = [(self.cheap_pre_score(a, game, color), a) for a in
                                              others]
1517            scored.sort(key=lambda x: x[0], reverse=True)
1518            top_k = [a for (_s, a) in scored[: self.PREFILTER_TOP_K]]
1519
1520            # Combine unique musts + top_k preserving order
1521            candidates = []
1522            for a in musts + top_k:
1523                if a not in candidates:
1524                    candidates.append(a)
1525
1526            # Fill up with random remaining samples until MAX_SIMULATIONS
1527            remaining = [a for a in all_actions if a not in candidates]
1528            random.shuffle(remaining)
1529            while len(candidates) < min(len(all_actions), self.
                                          MAX_SIMULATIONS) and
                                          remaining:
1530                candidates.append(remaining.pop())
1531
1532            if not candidates and all_actions:
1533                candidates = random.sample(all_actions, min(len(all_actions),
                                              self.MAX_SIMULATIONS))
1534
1535            self.debug_print(f"FooPlayer: Prefilter selected {len(candidates)
                                          } candidates (musts={len(
                                          musts)}, phase={phase})")
1537            if self.DEBUG and phase == "EARLY":
1538                self.debug_print(f"  Forced includes: settlement={'yes' if
                                              found_settlement else '
                                              no'}, road={'yes' if
                                              found_road else 'no'}")
1542            return candidates
1543
1544        # ------------------- Playable actions extraction -------------------
1545        def get_playable_actions_from_game(self, game: Game) -> List[Any]:
1546            """Try adapters.list_prunned_actions first, then common game
                                              attributes."""
1547            try:
1548                acts = list_prunned_actions(game)
1549                if acts:
1550                    return acts
1551            except Exception as e:
1552                self.debug_print("FooPlayer: list_prunned_actions unavailable
                                              or failed. Error:", e)
1553            try:
1554                if hasattr(game, "get_playable_actions"):
1555                    return list(game.get_playable_actions())
1556            except Exception:
1557                pass
1558            try:
1559                if hasattr(game, "playable_actions"):
1560                    return list(getattr(game, "playable_actions"))
1561            except Exception:
1562                pass
1563            try:
1564                state = getattr(game, "state", None)
1565                if state is not None and hasattr(state, "playable_actions"):
                    return list(getattr(state, "playable_actions"))
                except Exception:
                    pass
```

```
1566
1567            return []
1568
1569        # ------------------ Robber / Knight evaluation ------------------
1570        def evaluate_robber_action(self, action: Any, game: Game, color:
                                             Color) -> float:
1571            """Estimate the value of moving the robber (best-effort).
1572
1573            If the action does not specify a target hex, evaluate all hexes
1574                                          and prefer the
1575            one that maximizes opponent production loss.
1576            """
1577            score = 0.0
1578            try:
1579                # Base preference to include robber moves (use HIGH base for
                                                 aggressive play)
1580                score += self.ROBBER_BASE_SCORE_HIGH
1581                name = self._safe_action_name(action)
1582                # Try to parse a target hex id
1583                digits = [int(tok) for tok in name.split() if tok.isdigit()]
                   target = digits[0] if digits else None
1584
1585                # Die probabilities
1586                die_prob = {2: 1 / 36, 3: 2 / 36, 4: 3 / 36, 5: 4 / 36, 6: 5
                                                 / 36, 8: 5 / 36, 9: 4 /
1587                                                 36, 10: 3 / 36, 11: 2 /
1588                                                 36, 12: 1 / 36}
1589
1590                state = getattr(game, "state", game)
1591                board = getattr(state, "board", None) or getattr(game, "board
                                                 ", None)
1592                hexes = getattr(board, "hexes", None) or getattr(board, "
1593                                                 tiles", None) or []
1594
1595                # Map hex identifier to object (best-effort: use index or id)
1596                hex_map = {}
1597                for idx, h in enumerate(hexes):
1598                    try:
1599                        hid = getattr(h, "id", None) or getattr(h, "index",
                                                         None) or idx
1600                    except Exception:
1601                        hid = idx
1602                    try:
1603                        key = int(hid) if isinstance(hid, int) or (isinstance
                                                         (hid, str) and
1604                                                         hid.isdigit())
1605                                                         else idx
1606                    except Exception:
1607                        key = idx
                       hex_map[key] = h
1608
1609                # Determine best target if none specified
1610                targets_to_consider = [target] if target in hex_map else list
1611                                                 (hex_map.keys())
1612                # Compute production loss on opponents per candidate target
1613                opponents = []
1614                players = getattr(state, "players", None) or getattr(game, "
1615                                                 players", None) or []
1616                my_color = color
1617                if isinstance(players, dict):
1618                    for k, p in players.items():
                           if k == my_color or getattr(p, "color", None) ==
1619                                                         my_color:
                               continue
```

```
1620                    opponents.append(p)
1621            else:
1622                for p in players:
1623                    if getattr(p, "color", None) == my_color:
1624                        continue
1625                    opponents.append(p)
1626
1627        best_loss = 0.0
1628        best_steal = 0.0
1629        best_hex = None
1630        resource_value = {"ore": 3.0, "metal": 3.0, "wheat": 3.0, "
1631                                        grain": 3.0, "brick": 2.
1632                                        0, "lumber": 2.0, "wood"
1633                                        : 2.0, "sheep": 2.0}
1634
1635        for t in targets_to_consider:
1636            try:
1637                if t not in hex_map:
1638                    continue
1639                h = hex_map[t]
1640                num = getattr(h, "roll", None) or getattr(h, "number"
1641                                                , None) or
1642                                                getattr(h, "
1643                                                value", None)
1644                try:
1645                    num = int(num)
1646                except Exception:
1647                    num = None
1648                prob = die_prob.get(num, 0)
1649                total_prod_loss = 0.0
1650                steal_expected = 0.0
1651                for opp in opponents:
1652
1653                    opp_settles = getattr(opp,
1654                    "settlements", None) or getattr(opp,
1655                    "settle_locations", None) or []
1656                    opp_cities = getattr(opp, "cities",
1657                    None) or getattr(opp, "city_locations",
1658                    None) or []
1659                    mult = 0.0
1660                    try:
1661                        for s in opp_settles:
1662                            neighbors = getattr(h,
1663                            "vertices", None) or getattr(h,
1664                            "adjacent_vertices", None) or []
1665                            if s in neighbors:
1666                                mult += 1.0
1667                        for c in opp_cities:
1668                            neighbors = getattr(h,
1669                            "vertices", None) or getattr(h,
1670                            "adjacent_vertices", None) or []
1671                            if c in neighbors:
1672                                mult += 2.0
1673                    except Exception:
                            continue
                    total_prod_loss += prob * mult
                    # Estimate steal expected
                    try:
                        opp_resources = getattr(opp,
                        "resources", None) or {}
                        if isinstance(opp_resources, dict)
                        and opp_resources:
                            total_res =
                            sum(opp_resources.values())
                            if total_res > 0:
```

```
1674                                    avg_val =
1675                                    sum(resource_value.get(r,
1676                                    1.5) * (opp_resources.get(r,
1677                                    0) / total_res) for r in
1678                                    opp_resources)
1679                                    steal_expected += avg_val *
                                        0.5
1680                        except Exception:
1681                            pass
1682                    # choose best
1683                    if total_prod_loss > best_loss or (abs(
1684                                                total_prod_loss
1685                                                - best_loss) <
1686                                                1e-9 and
                                                steal_expected >
1687                                                 best_steal):
1688                        best_loss = total_prod_loss
1689                        best_steal = steal_expected
1690                        best_hex = t
                except Exception:
1691                    continue
1692
1693            # Aggressive scaling per latest tuning
1694            score += best_loss * self.PROD_LOSS_IMPORTANCE
1695            score += best_steal * 30.0
1696            # Extra bonus if multiple opponent cities affected
            try:
1697                if best_hex in hex_map:
1698                    h = hex_map[best_hex]
1699                    city_count = 0
1700                    for opp in opponents:
1701                        for c in getattr(opp, "cities", []) or
                            getattr(opp, "city_locations", []) or []:
1702                            neighbors = getattr(h, "vertices",
1703                            None) or getattr(h,
1704                            "adjacent_vertices", None) or []
                            if c in neighbors:
1705                                city_count += 1
1706                    if city_count > 0:
1707                        score += 20.0 * city_count
            except Exception:
1708                pass
1709
1710            # If steal estimated is very significant, add
1711            decisive bonus
1712            if best_steal > 2.0:
1713                score += 30.0
1714
1715            # Debug
            if self.DEBUG and best_hex is not None:
1716                self.debug_print(f"FooPlayer: evaluate_robber_action
1717                                            best_hex={best_hex}
1718                                            prod_loss={best_loss
1719                                            :.3f} steal_ev={
                                            best_steal:.2f}")
1720
1721        except Exception:
1722            pass
1723        return float(score)
1724
1725    def evaluate_play_knight(self, action: Any, game: Game, color: Color)
                                        -> float:
1726        """Estimate the value of playing a knight (best-effort)."""
1727        score = float(self.KNIGHT_BASE)
        try:
```

```
1728            name = self._safe_action_name(action)
1729            if "steal" in name or "rob" in name:
1730                score += 10.0
1731
1732            # army progress
1733            player_state = self._get_player_state(game, color)
1734            army = getattr(player_state, "army", None) or getattr(
1735                                                player_state, "army_size
1736                                                ", None) or getattr(
1737                                                player_state, "
1738                                                knights_played", None)
1739                                                or 0
1740            try:
1741                army = int(army)
1742            except Exception:
1743                army = 0
1744
1745            # detect largest army threshold
1746            largest_threshold = 3
1747            try:
1748                state = getattr(game, "state", game)
1749                players = getattr(state, "players", None) or getattr(game
1750                                                , "players", None)
1751                                                or []
1752                max_other = 0
1753                if isinstance(players, dict):
1754                    for k, p in players.items():
1755                        if getattr(p, "color", None) == color or k ==
1756                                                            color:
1757                            continue
1758                        other_army = getattr(p, "army", None) or getattr(
1759                                                    p, "
1760                                                    army_size",
1761                                                    None) or
1762                                                    getattr(p, "
1763                                                    knights_played
1764                                                    ", None) or
1765                                                    0
1766                        try:
1767                            other_army = int(other_army)
1768                        except Exception:
1769                            other_army = 0
1770                        max_other = max(max_other, other_army)
1771                else:
1772                    for p in players:
1773                        if getattr(p, "color", None) == color:
1774                            continue
1775                        other_army = getattr(p, "army", None) or getattr(
1776                                                    p, "
1777                                                    army_size",
1778                                                    None) or
1779                                                    getattr(p, "
1780                                                    knights_played
1781                                                    ", None) or
                                                    0
                        try:
                            other_army = int(other_army)
                        except Exception:
                            other_army = 0
                        max_other = max(max_other, other_army)
                largest_threshold = max(3, max_other + 1)
            except Exception:
                largest_threshold = 3

            if army + 1 >= largest_threshold:
```

```python
                    score += self.KNIGHT_LARGEST_ARMY_BONUS
                else:
                    score += 20.0

                # Debug
                if self.DEBUG:
                    self.debug_print(f"FooPlayer: evaluate_play_knight army={
                                                army} target={
                                                largest_threshold}
                                                score={score}")

        except Exception:
            pass
        return float(score)

    # ------------------- Helper: determine active player color
                        -------------------
    def _get_active_player_color(self, game: Game) -> Optional[Color]:
        """Best-effort to detect which Color is to move in the given game
                                    state."""
        try:
            state = getattr(game, "state", game)
            cp = getattr(state, "current_player", None) or getattr(state,
                                            "active_player", None)
                                            or getattr(state, "
                                            turn_color", None)
            if cp is None:
                cp = getattr(game, "current_player", None)
            # cp might be index, player object, or Color
            if isinstance(cp, Color):
                return cp
            if isinstance(cp, int):
                players = getattr(state, "players", None) or getattr(game
                                            , "players", None)
                                            or []
                try:
                    if isinstance(players, (list, tuple)) and 0 <= cp <
                                                len(players):
                        return getattr(players[cp], "color", None)
                except Exception:
                    pass
            # If cp is a player object
            if hasattr(cp, "color"):
                return getattr(cp, "color")

            # Fallback: pick first player in players whose color != our
                                            color
            players = getattr(state, "players", None) or getattr(game, "
                                            players", None) or []
            my_color = self._get_player_color()
            if isinstance(players, dict):
                for k, p in players.items():
                    try:
                        c = getattr(p, "color", None) or k
                        if c != my_color:
                            return c
                    except Exception:
                        continue
            else:
                for p in players:
                    try:
                        c = getattr(p, "color", None)
                        if c != my_color:
                            return c
                    except Exception:
                        continue
```

```python
        except Exception:
            pass
        return None

    # ------------------- Rollout logic with opponent-response
                                            -------------------
    def rollout_value(self, game: Game, color: Color, depth: int, initial
                                            : bool = True) -> float:
        """Short greedy rollout with phase bias and light opponent-
                                            response.

        initial: True for the first step of rollout so we can bias toward
                                            expansion early.
        """
        try:
            if depth <= 0:
                return self._evaluate_game_state(game, color)

            actions = self.get_playable_actions_from_game(game)
            if not actions:
                return self._evaluate_game_state(game, color)

            phase = self.get_game_phase(game, color)

            def score_for_rollout(a, g, c, is_initial):
                base = self.cheap_pre_score(a, g, c)
                if is_initial and phase == "EARLY":
                    name = self._safe_action_name(a)
                    if any(tok in name for tok in ("build_settlement", "
                                            build_sett", "
                                            settle")):
                        base *= self.ROLLOUT_SETTLEMENT_BONUS
                    if any(tok in name for tok in ("build_road", "road"))
                                            :
                        base *= self.ROLLOUT_ROAD_BONUS
                return base

            sorted_actions = sorted(actions, key=lambda a:
                                            score_for_rollout(a,
                                            game, color, initial),
                                            reverse=True)

            # Try top actions to simulate
            for a in sorted_actions[:6]:
                branches = []
                try:
                    branches = execute_deterministic(game, a)
                except Exception:
                    try:
                        branches = execute_spectrum(game, a)
                    except Exception:
                        branches = []
                if not branches:
                    continue
                # pick the most probable branch
                next_game = max(branches, key=lambda bp: float(bp[1]))[0]

                # Light opponent-response: if opponent to move next,
                                            simulate their
                                            greedy action once
                opp_color = self._get_active_player_color(next_game)
                my_color = color
                if opp_color is not None and opp_color != my_color and
                                            depth >= 2:
                    try:
```

```
                            opp_actions = self.get_playable_actions_from_game
                                                              (next_game)
                        if opp_actions:
                            # filter out robber/knight for
                            opponent response unless all are
                            robber/knight
                            non_disrupt = [oa for oa in
                            opp_actions if not any(tok in
                            self._safe_action_name(oa) for tok
                            in ("knight", "robber",
                            "move_robber"))]
                            candidate_ops = non_disrupt if
                            non_disrupt else opp_actions
                            # pick opponent best action by
                            cheap_pre_score from their
                            perspective
                            best_opp = max(candidate_ops,
                            key=lambda oa:
                            self.cheap_pre_score(oa, next_game,
                            opp_color))
                            # simulate opponent action
                            deterministically if possible
                            opp_branches = []
                            try:
                                opp_branches =
                                execute_deterministic(next_game,
                                best_opp)
                            except Exception:
                                try:
                                    opp_branches =
                                    execute_spectrum(next_game,
                                    best_opp)
                                except Exception:
                                    opp_branches = []
                            if opp_branches:
                                next_game = max(opp_branches,
                                key=lambda bp: float(bp[1]))[0]
                    except Exception:
                        pass

            return self.rollout_value(next_game, color,
            depth - 1, initial=False)

        # fallback: try any action that simulates
        for a in sorted_actions[:10]:
            branches = []
            try:
                branches = execute_deterministic(game, a)
            except Exception:
                try:
                    branches = execute_spectrum(game, a)
                except Exception:
                    branches = []
            if branches:
                next_game = max(branches, key=lambda bp:
                float(bp[1]))[0]
                return self.rollout_value(next_game, color,
                depth - 1, initial=False)

        return self._evaluate_game_state(game, color)
    except Exception as e:
        self.debug_print("FooPlayer: rollout_value exception, falling
                                      back to
                                      evaluate_game_state.
                                      Error:", e)
```

```
1944                return self._evaluate_game_state(game, color)
1945
1946        # ------------------ Evaluate action expectation (enhanced)
1947                                      ------------------
1948        def _evaluate_action_expectation(self, game: Game, action: Any,
1949                                        per_action_branch_limit: int = 8
1950                                        ) -> float:
1951            """Compute expected value of taking `action` in `game` for this
1952                                          player.
1953            Uses execute_spectrum when available then adds a rollout estimate
1954                                          for depth-1.
1955            """
1956            color = self._get_player_color()
1957            # Quick boosts for robber/knight/dev before heavy sim
1958            name = self._safe_action_name(action)
1959            preboost = 0.0
1960            try:
1961                if any(tok in name for tok in ("move_robber", "robber")):
1962                    preboost += self.evaluate_robber_action(action, game,
1963                                                color)
1964                if any(tok in name for tok in ("knight", "play_knight")):
1965                    preboost += self.evaluate_play_knight(action, game, color
1966                                                )
1967                if any(tok in name for tok in ("buy_dev", "buycard", "
1968                                                buy_dev_card")):
1969                    try:
1970                        dev_ev = self.dev_card_ev_estimate(game, color)
1971                        preboost += dev_ev * self.DEV_EV_SCALE
1972                    except Exception:
1973                        # fallback small preboost
1974                        preboost += 20.0
1975            except Exception:
1976                preboost += 0.0
1977
1978            branches = None
1979            try:
1980                branches = execute_spectrum(game, action)
1981                if not branches:
1982                    raise RuntimeError("execute_spectrum returned no branches
1983                                        ")
1984            except Exception as e_s:
1985                self.debug_print("FooPlayer: execute_spectrum failed or
1986                                            unavailable for action;
1987                                            trying deterministic.
1988                                            Error:", e_s)
1989                try:
1990                    branches = execute_deterministic(game, action)
1991                    if not branches:
1992                        raise RuntimeError("execute_deterministic returned no
1993                                            outcomes")
1994                except Exception as e_d:
1995                    self.debug_print("FooPlayer: Both execute_spectrum and
1996                                                execute_deterministic
1997                                                failed for action.
                                                Errors:", e_s, e_d)
                    return float("-inf")

            # Limit branches to keep runtime bounded
            if len(branches) > per_action_branch_limit:
                branches = sorted(branches, key=lambda bp: float(bp[1]),
                                            reverse=True)[:
                                            per_action_branch_limit]
```

```
1998            expected = 0.0
1999            total_prob = 0.0
2000            rollout_depth = max(0, self.ROLLOUT_DEPTH - 1)
2001            for (out_game, prob) in branches:
2002                try:
2003                    # For buy_dev actions, if the branch encodes a known draw
                                                     outcome, we could
2004                                                 refine.
2005                    # In absence of explicit draw info, rely on
2006                                                 dev_ev_estimate as a
2007                                                 conservative proxy.
2008                    immediate = self._evaluate_game_state(out_game, color)
2009                    rollout_est = self.rollout_value(out_game, color,
2010                                                 rollout_depth,
2011                                                 initial=True)
2011                    branch_val = 0.6 * immediate + 0.4 * rollout_est
2012                except Exception as e:
2013                    self.debug_print("FooPlayer: evaluation failed for branch
                                                 , using heuristic.
2014                                                 Error:", e)
2015                    branch_val = self._heuristic_value(out_game, color)
2016                expected += float(prob) * float(branch_val)
2017                total_prob += float(prob)

2018            if total_prob > 0:
2019                expected = expected / total_prob
2020
2021            expected += preboost
2022            return float(expected)

2023        # ------------------ Main decision function -------------------
2024        def decide(self, game: Game, playable_actions: Iterable) -> Optional[
2025                                          object]:
2026            """Choose an action from playable_actions using phase-aware
2027                                          sampling + rollouts."""
2028            try:
2028                playable_actions = list(playable_actions)
2029                if not playable_actions:
2030                    self.debug_print("FooPlayer: No playable actions
2031                                                 available, returning
2032                                                 None")
2033                    return None

2034                color = self._get_player_color()
2035                phase = self.get_game_phase(game, color)

2036                # Prefilter candidate actions
2037                candidates = self.prefilter_actions(playable_actions, game,
2038                                                 color)

2040                # Cap to MAX_SIMULATIONS
2041                if len(candidates) > self.MAX_SIMULATIONS:
2042                    candidates = candidates[: self.MAX_SIMULATIONS]

2043                if not candidates:
2044                    candidates = random.sample(playable_actions,
2045                    min(len(playable_actions), self.MAX_SIMULATIONS))

2047                # Distribute simulation budget adaptively
2048                per_action_budget = max(1, self.SIMULATION_BUDGET //
2048                max(1, len(candidates)))

2050                best_score = float("-inf")
2051                best_actions: List[Any] = []
                scores_debug: List[Tuple[float, Any]] = []
```

```
2052
2053            for a in candidates:
2054                try:
2055                    score =
2056                    self._evaluate_action_expectation(game, a,
                        per_action_branch_limit=per_action_budget)
2057                except Exception as e:
2058                    self.debug_print("FooPlayer: Exception
2059                    during action evaluation, skipping action.
2060                    Error:", e)
2061                    score = float("-inf")
2062
2063                scores_debug.append((score, a))
2064
                    if score > best_score + self.TOLERANCE:
2065                    best_score = score
2066                    best_actions = [a]
2067                elif abs(score - best_score) <= self.TOLERANCE:
                    best_actions.append(a)
2068
2069            # If no action had a finite score, fallback to first playable
2070                                                     action
2071            if not best_actions:
2072                self.debug_print("FooPlayer: All evaluations failed,
2073                                                 defaulting to first
                                                 playable action")
2074                return playable_actions[0]
2075
2076            # Epsilon-greedy randomness to reduce predictability
2077            chosen: Any
                scores_debug.sort(key=lambda x: x[0], reverse=True)
2078            if random.random() < self.EPSILON_GREEDY and len(scores_debug
2079                                             ) >= 2:
2080                # pick from top-3 weighted by score (or fewer if not
2081                                                 available)
2082                top_k = scores_debug[: min(3, len(scores_debug))]
                weights = [max(0.0, s - top_k[-1][0] + 1e-6) for (s, a)
2083                                                 in top_k]
2084                total_w = sum(weights)
2085                if total_w > 0:
2086                    r = random.random() * total_w
2087                    cum = 0.0
2088                    for w, (_s, a) in zip(weights, top_k):
2089                        cum += w
                        if r <= cum:
2090                            chosen = a
2091                            break
2092                    else:
2093                        chosen = top_k[0][1]
                else:
2094                    chosen = scores_debug[0][1]
2095                if self.DEBUG:
2096                    self.debug_print(f"FooPlayer: EPSILON pick
2097                    triggered, chosen alternate action {chosen}")
                return chosen
2098
2099            # If tie, break ties preferring
2100            settlement/road/resource diversity improvements
2101            if len(best_actions) > 1:
2102                tie_metrics = []
2103                for a in best_actions:
2104                    try:
                        metric = 0.0
2105                        metric += self.settlement_potential(a,
```

```
2106                        game, color)
2107                        metric +=
2108                        self.road_connection_potential(a, game,
2109                        color)
2110                        # small production proxy via heuristic
2111                        metric += 0.01 *
2112                        self._heuristic_value(game, color)
2113                        tie_metrics.append((metric, a))
2114                    except Exception:
2115                        tie_metrics.append((0.0, a))
2116                tie_metrics.sort(key=lambda x: x[0],
2117                reverse=True)
2118                # pick the top metric actions (could still be
2119                multiple)
2120                top_metric = tie_metrics[0][0]
2121                filtered = [a for (m, a) in tie_metrics if abs(m
2122                - top_metric) <= self.TOLERANCE]
2123                if filtered:
2124                    chosen = random.choice(filtered)
2125                else:
2126                    chosen = random.choice(best_actions)
2127            else:
2128                chosen = best_actions[0]

            # Debug logging: phase and top candidates
            if self.DEBUG:
                self.debug_print(f"FooPlayer: Phase={phase},
                SettlementsTarget=
                {self.TARGET_SETTLEMENTS_EARLY}")
                topn = scores_debug[:3]
                self.debug_print("FooPlayer: Top candidates:")
                for sc, act in topn:
                    self.debug_print(f"  score={sc:.2f} action=
                    {act}")

            self.debug_print(f"FooPlayer: Chosen action {chosen}
            with expected score {best_score}")
            return chosen
        except Exception as e:
            # Protect against unexpected errors
            print("FooPlayer: Unexpected error in decide(),
            defaulting to first playable action. Error:", e)
            try:
                return list(playable_actions)[0]
            except Exception:
                return None
```

## A.2 HexMachina (No Discovery) Shallow Strategy

```
import random
import statistics
from .adapters import (
    Game, Player, Color
)

# Hyperparameters for this player. Tweak across evolutions.
K_ROLLOUTS = 0  # rollouts disabled in this adapter-limited
                                    implementation
MAX_ROLLOUT_DEPTH = 10  # not used currently; kept for future use
MAX_ACTIONS_TO_EVALUATE = 12
DEBUG = True
```

```python
class FooPlayer(Player):
    """A stronger FooPlayer that performs a 1-ply lookahead and evaluates
    the immediate successor state using a robust, defensive static
                                    evaluator.

    Notes on integration with adapters.py:
    - We only use the thin adapter surface exported above (Game, Player,
                                    Color).
    - We call game.copy() to create hypothetical states and game.execute(
                                    action)
      to apply actions to those copies. We avoid calling any non-exported
      adapter helpers so this file remains compatible with the framework.

    Limitations and rationale:
    - The adapters surface available in this environment does not
                                    explicitly
      expose helper functions for enumerating playable actions from an
      arbitrary game object (those are provided to decide() by the
                                    harness).
      Because of this we cannot reliably perform multi-step random
                                    rollouts
      (we cannot ask the engine for "playable_actions" inside the player
                                    for
      subsequent turns). Attempting to call hypothetical internal APIs
                                    would
      risk using non-portable / unsupported functions.
    - To still fix the key flaw (always pick the first action) we
                                    implement a
      1-ply lookahead over a sampled set of candidate actions and
                                    evaluate the
      successor state with a robust static value function that inspects
                                    the
      game.state. This is a significant upgrade over the previous
                                    behavior
      and provides a solid foundation for future rollout-based evolution.
    """

    def __init__(self, name=None):
        super().__init__(Color.BLUE, name)

    def decide(self, game, playable_actions):
        """Choose an action from playable_actions.

        Strategy implemented:
        - If there are many playable actions, randomly sample up to
          MAX_ACTIONS_TO_EVALUATE actions to limit computation.
        - For each candidate action, copy the game, execute the action on
                                        the
          copy, and evaluate the resulting state with _evaluate_state().
        - Choose the action with the highest evaluation. Break ties
                                        randomly.

        The evaluation is defensive: it attempts multiple common access
        patterns to extract victory points and common counts (settlements
                                        ,
        cities, roads). If extraction fails, the evaluator falls back to
                                        0.

        Args:
            game (Game): complete game state. read-only. Use game.copy()
                                            to
                        create hypothetical states.
            playable_actions (Iterable[Action]): legal options for this
                                            turn.
```

```
2214          Returns:
2215              action: chosen element of playable_actions, or None if no
2216                                                 options.
2217          """
2218          # Defensive: if no actions available, return None
2219          if not playable_actions:
2220              if DEBUG:
2221                  print('FooPlayer.decide: no playable_actions -> returning
2222                                                  None')
2223              return None
2224          # Convert playable_actions to a list so we can sample and index
2225          try:
2226              actions = list(playable_actions)
2227          except Exception:
2228              # If iterable cannot be converted, fall back to returning
2229                                                  first
2230              if DEBUG:
2231                  print('FooPlayer.decide: playable_actions not list-like;
2232                                                      defaulting to first'
2233                                                      )
2234              try:
2235                  return playable_actions[0]
2236              except Exception:
2237                  return None

2238          # Sample candidate actions if there are too many
2239          if len(actions) > MAX_ACTIONS_TO_EVALUATE:
2240              candidates = random.sample(actions, MAX_ACTIONS_TO_EVALUATE)
2241              if DEBUG:
2242                  print(f'FooPlayer.decide: sampled {len(candidates)} of {
2243                                                      len(actions)}
2244                                                      actions to evaluate'
2245                                                      )
2246          else:
2247              candidates = actions
2248              if DEBUG:
2249                  print(f'FooPlayer.decide: evaluating all {len(candidates)
2250                                                      } actions')

2251          # Evaluate each candidate action by applying it to a copy of the
2252                                                  game
2253          scores = []  # list of (action, score)
2254          for i, action in enumerate(candidates):
2255              try:
2256                  # Copy the game to avoid mutating the original
2257                  new_game = game.copy()

2258                  # Apply the candidate action on the copied game.
2259                  # The standard Game API exposes execute(action) to apply
2260                                                      an action.
2261                  # We try both .execute and .apply for defensive
2262                                                      compatibility.
2263                  executed = False
2264                  try:
2265                      new_game.execute(action)
2266                      executed = True
2267                  except Exception:
2268                      # Some versions may expose a differently named method
2269                                                      .
2270                      try:
2271                          new_game.apply(action)
2272                          executed = True
2273                      except Exception:
2274                          executed = False
```

```
2268
2269                if not executed:
2270                    # If we couldn't apply the action on the copy, mark
                                                        it as
2271
2272                    # very poor and continue.
                    if DEBUG:
2273
                        print(f'FooPlayer.decide: failed to execute
2274
                                                        candidate
2275                                                    action {i};
2276                                                    marking
2277                                                    score -inf')
                    scores.append((action, float('-inf')))
2278
                    continue
2279
2280                # Evaluate the successor state
2281                score = self._evaluate_state(new_game)
2282                scores.append((action, score))
                    if DEBUG:
2283
                        print(f'FooPlayer.decide: action #{i} -> score {score
2284                                                    }')
2285
2286        except Exception as e:
                    # Catch-all: do not let the player crash the harness.
2287
                                                        Penalize
2288                # the action and continue evaluating others.
                    if DEBUG:
2289
2290                        print(f'FooPlayer.decide: exception while evaluating
2291                                                    action #{i}: {e}
2292                                                    ! Marking -inf')
                    scores.append((action, float('-inf')))
2293
2294        # Choose the best action. If all are -inf or evaluation failed,
                                                fall back
2295
2296        # to the original first-action policy.
        if not scores:
2297
            if DEBUG:
2298
                print('FooPlayer.decide: no scores produced -> defaulting
2299                                                to first action')
2300            return actions[0]
2301
2302        # Compute the maximum score
        max_score = max(score for (_, score) in scores)
2303
        # Filter all actions that have the max score (handle ties)
2304        best_candidates = [a for (a, s) in scores if s == max_score]
2305
2306        if not best_candidates or max_score == float('-inf'):
            # All evaluations failed; fallback
2307
            if DEBUG:
2308
                print('FooPlayer.decide: all evaluations failed ->
2309                                                defaulting to first
2310                                                action')
2311            return actions[0]
2312
2313        chosen = random.choice(best_candidates)
        if DEBUG:
2314
            try:
2315
                # Try to pretty-print a small summary for debugging
2316                print(f'FooPlayer.decide: selected action -> {repr(chosen
2317                                                )} with score {
                                                max_score}')
2318
            except Exception:
2319
                print('FooPlayer.decide: selected an action (repr failed)
2320                                                ')
2321
        return chosen
```

```python
def _evaluate_state(self, game):
    """Static evaluation of a game state from this player's
                                    perspective.

    The evaluator attempts multiple common access patterns to extract
    victory points and simple progress indicators (settlements,
                                    cities,
    roads). The returned score is primarily the victory points (
                                    higher is
    better). Secondary counts are used as small tiebreakers.

    This function is defensive to avoid attribute errors across
                                    different
    engine versions.

    Returns:
        float: heuristic score for the state (larger is better)
    """
    color = self.color
    vp = None
    settlements = None
    cities = None
    roads = None

    # Try a number of plausible attribute access patterns. Use try/
                                    except
    # blocks liberally because different engine versions expose
                                    different
    # structures.
    try:
        players = game.state.players
    except Exception:
        players = None

    # Attempt to access player state by Color key
    player_state = None
    if players is not None:
        try:
            player_state = players[color]
        except Exception:
            # Maybe players is a list keyed by integer colors
            try:
                idx = int(color)
                player_state = players[idx]
            except Exception:
                player_state = None

    # Extract victory points with common attribute names
    if player_state is not None:
        for attr in ('victory_points', 'victoryPoints', 'vp', 'points
                                    '):
            try:
                val = getattr(player_state, attr)
                # If it's a callable (method), call it
                if callable(val):
                    val = val()
                vp = int(val)
                break
            except Exception:
                vp = None

        # Try dictionary-style if attributes failed
        if vp is None:
            try:
```

```
2376                    if isinstance(player_state, dict):
2377                        for key in ('victory_points', 'vp', 'points'):
2378                            if key in player_state:
2379                                vp = int(player_state[key])
2380                                break
2381                except Exception:
2382                    vp = None

2383            # Extract simple asset counts to break ties
2384            for attr in ('settlements', 'settle_count', 'settlement_count
2385                                                ', 'settles'):
2386                try:
2387                    val = getattr(player_state, attr)
2388                    if callable(val):
2389                        val = val()
2390                    settlements = int(val)
2391                    break
2392                except Exception:
2393                    settlements = None

2393            for attr in ('cities', 'city_count'):
2394                try:
2395                    val = getattr(player_state, attr)
2396                    if callable(val):
2397                        val = val()
2398                    cities = int(val)
2399                    break
2400                except Exception:
2401                    cities = None

2401            for attr in ('roads', 'road_count'):
2402                try:
2403                    val = getattr(player_state, attr)
2404                    if callable(val):
2405                        val = val()
2406                    roads = int(val)
2407                    break
2408                except Exception:
2409                    roads = None

2410        # Fallbacks if extraction failed: try to compute from visible
2411                                        board pieces
2412        # (e.g., lengths of lists). This is optional and best-effort.
2412        if vp is None and players is not None:
2413            try:
2414                # If player_state contains lists of pieces, inspect
2415                                                lengths
2416                if isinstance(player_state, dict):
2417                    # Look for settlement/city lists
2418                    s = None
2418                    for key in ('settlements', 'settle_list'):
2419                        if key in player_state and isinstance(
2420                                                        player_state
2421                                                        [key], (list
2421                                                        , tuple)):
2422                            s = len(player_state[key])
2423                            break
2424                    if s is not None:
2425                        settlements = settlements or s
2426                # We intentionally do not try to derive vp from the board
2427                                                in a
2427                # brittle way; leave vp as None and fall back to 0.
2428            except Exception:
2429                pass
```

```
        # Final fallback: if we couldn't determine vp, set to 0
        if vp is None:
            vp = 0

        # Build a composite score. Main contributor is victory points.
                                        Add
        # small weighted bonuses for settlements/cities/roads if
                                        available.
        score = float(vp)
        if settlements is not None:
            score += 0.01 * float(settlements)
        if cities is not None:
            score += 0.02 * float(cities)
        if roads is not None:
            score += 0.005 * float(roads)

        return score
```

