# OpenReview forum: "HexMachina: Self-Evolving Multi-Agent System for Continual Learning of Catan"
_ICLR.cc/2026/Conference — ICLR 2026 Conference Withdrawn Submission_

### Official Review · Reviewer_pNbf · 2025-10-29

**Soundness:** 3
**Presentation:** 3
**Contribution:** 2
**Rating:** 2
**Confidence:** 4

**Summary:**

The paper introduced HexMachina, an agentic game-playing framework that receives feedbacks from gameplay and then evolve a python code as policy. The paper tests its method in the game called Settlers of Catan, and the results show that the method leads to better policy after rounds of evolution.

**Strengths:**

1. The proposed method is sound and the performance beats the baselines compared in the paper.

**Weaknesses:**

1. Lack of novelty. Receiving feedbacks from game environments and leveraging them to improve python-code-as-policy is not a new idea. And the way to implement it using role-playing is also a common practice.
2. Lack of in-depth analysis. For example, as mentioned in the last paragraph, current method is limited by context length. However, it is mentioned or analyzed in the paper.
3. Lack of baselines. The baselines seem to be simple. Is it possible to include more baselines? For example, is it possible to train an agent using RL as a learning-based baseline?

**Questions:**

1. The alphabeta baseline seems to perform quite well already (according to Table 2). Given a larger budget, would it surpass the FooPlayer?
2. Why are the experiments conducted solely in a two-player setting, given that the paper introduced it as a 3–4 player game? It would be interesting to see how the agent evolves in a multi-player game environment.

---

### Official Review · Reviewer_XLCZ · 2025-10-30

**Soundness:** 2
**Presentation:** 2
**Contribution:** 2
**Rating:** 2
**Confidence:** 3

**Summary:**

The paper proposes HexMachina, a self-evolving multi-agent system on Catan games, which separates environment discovery and strategy improvement, so it can focus on high-level strategy design rather than per-turn decision-making. In order to verify the system, the paper conducts experiments on Catan games by comparing several baselines including Random, LLM Player, basic continual player and AlphaBeta.

**Strengths:**

The paper proposes a self-evolving multi-agent system for Catan games, which separates environment discovery and strategy improvement. This design can evolve players that consistently execute intelligent, long-horizon strategies, outperforming traditional LLM agents.

The paper conducts experiments on Catan games, and outperforms several baselines, such as prompt-driven baselines and alphabeta.

**Weaknesses:**

The paper is hard to follow, and many parts lack clarity. For example: How does the multi-agent system actually operate and evolve? how to learn an unknown environment without any human documentation, especially inducing execute code into Adapter files? how to preserve key code and knowledge as artifacts, and improve its strategy via a closed-loop process? Why is the system able to consistently carry out a long-term plan across an entire game? How exactly does each agent (Orchestrator, Analyst, Coder, Researcher, and Strategist) perform its tasks? How does the Memory mechanism function in practice?

Additionally, a significant limitation is that all experiments are conducted in a single environment (Catan). The generalizability of the approach to other long-horizon, adversarial domains remains unverified. The paper also does not compare against other continual LLM agents (e.g., Voyager, Eureka) in the same environment.

While the architectural components of HexMachina, such as multi-agent LLM systems and code generation for policy implementation, are well-established in prior works like GameGPT, MetaGPT, CAMEL, and Voyager, the paper should better articulate what is fundamentally new in HexMachina beyond the application to Catan.

**Questions:**

See the weaknesses.

---

### Official Review · Reviewer_8Ma2 · 2025-10-31

**Soundness:** 2
**Presentation:** 2
**Contribution:** 3
**Rating:** 4
**Confidence:** 3

**Summary:**

This paper introduces HexMachina, a self-evolving multi-agent system that enables large language models to learn long-horizon strategies in Settlers of Catan. By separating environment discovery from strategy improvement, the system allows the LLM to generate and refine executable code autonomously. Experiments show a 54% win rate against the AlphaBeta baseline, surpassing prompt-based agents. While the idea of artifact-centric continual learning is promising, the results are limited to one domain and modest in scope.

**Strengths:**

- The paper addresses an important challenge—long-horizon strategy learning for LLM agents—by introducing a novel self-evolving multi-agent system.
- The separation of environment discovery and strategy improvement is an original and well-motivated design choice that enhances coherence and performance.
- The work is clearly presented and empirically validated on a meaningful benchmark (Settlers of Catan), demonstrating measurable gains over strong baselines.

**Weaknesses:**

1. Limited empirical scope: Experiments are confined to a single domain (Settlers of Catan), which weakens claims about general continual learning or long-horizon reasoning. Broader validation across different environments would strengthen the argument.
2. Marginal improvement over baselines: While the win rate exceeds AlphaBeta slightly, it is unclear whether the gain reflects genuine strategic learning or task-specific heuristics. More analysis of learned behaviors is needed.
3. High system complexity and cost: The multi-agent setup requires multiple LLMs and heavy computation, raising concerns about scalability and practicality for broader deployment.
4. Insufficient system visualization and excessive appendix code: The paper lacks a clear architecture diagram and illustrative case study that explain how the components interact throughout the self-evolution process, while the appendix includes excessive raw code that detracts from clarity and readability.

**Questions:**

1. How generalizable is HexMachina to other environments beyond Catan? Could the same discovery–improvement architecture transfer to non-game domains (e.g., robotics or programming tasks)?
2. What concrete mechanisms ensure stability during self-evolution (e.g., avoiding catastrophic forgetting or overfitting to AlphaBeta’s style)?
3. How are adapter induction and code validation automated? Are there failure cases where the induced API abstraction was incorrect or inconsistent?
4. What is the computational cost (in tokens or hours) per evolution step, and how does it scale with larger LLMs?
5. Were any qualitative or behavioral analyses performed on the evolved agents to verify that HexMachina truly develops long-horizon planning behaviors—such as consistent expansion strategies, adaptive trading, or opponent modeling—rather than merely optimizing short-term heuristics for higher win rates?
6. Is there a risk that HexMachina simply overfits to the fixed AlphaBeta opponent rather than learning generally robust strategies?
7. How dependent is the framework on high-end proprietary models (e.g., GPT-5-mini, Claude 3.7)? Could comparable performance be achieved with open-source LLMs such as Llama 3 or Qwen2.5, or would performance degrade substantially?

---

### Official Review · Reviewer_dBFM · 2025-10-31

**Soundness:** 2
**Presentation:** 2
**Contribution:** 2
**Rating:** 0
**Confidence:** 3

**Summary:**

HexMachina is a multi-agent system that learns while it plays by having several different agents with different roles interact to refine their game strategy. An orchestrator agent during the discovery phase tasks other agents to write code in a adapters file that can be executed in the catan game. Then during the improvement phase the agents are orchestrated to improve the code that will be run during the game and refine it as it plays.

**Strengths:**

- The work shows some really interesting results when the system learns continually as it plays Catan, evetually beating a strong human policy baseline.
- Dual memory system is interesting, comprising of the game memory and semantic memory to enable continual learning.

**Weaknesses:**

- Only a single game is tested, Settlers of Catan. It would improve the work much more if more games were considered, otherwise I find that it is easy to make erroneous conclusions from just results on a single noisy game.
- The ablations in 6.3 seem too superficial. I was interested in seeing whether the multi-agent setup really mattered or if a single agent would also work fine. While 6.3 ablations remove different agents from the setup, its unclear how they are removed and how the fewer agent setup is designed. What are the prompts, what tools do they have or are aware of etc.
- Limitation section mentions that code hallucination was a common failure case and had to be addressed. how was this addressed, did the system simply iterate on the code until the code ran without errors, or was there human intervention? This is quite unclear.
- Appendix lacks significant information, e.g. what are the prompts each agent is given. With the lack of information, it becomes more unclear how such a multi-agent system can be reproduced or whether it even matters.

**Questions:**

See above.

---

### Note · Authors · 2025-12-01

I have read and agree with the venue's withdrawal policy on behalf of myself and my co-authors.